# Vertebrate host phylogeny influences gut archaeal diversity

Nicholas D. Youngblut [1] ✉, Georg H. Reischer[2,3], Silke Dauser[1], Sophie Maisch[1], Chris Walzer [4,5], Gabrielle Stalder[5], Andreas H. Farnleitner[2,3,6] and Ruth E. Ley[1,7]

Commonly used 16S rRNA gene primers do not detect the full range of archaeal diversity present in the vertebrate gut. As a result, several questions regarding the archaeal component of the gut microbiota remain, including which Archaea are host-associated, the specificities of such associations and the major factors influencing archaeal diversity. Using 16S rRNA gene amplicon sequencing with primers that specifically target Archaea, we obtained sufficient sequence data from 185 gastrointestinal samples collected from 110 vertebrate species that span five taxonomic classes (Mammalia, Aves, Reptilia, Amphibia and Actinopterygii), of which the majority were wild. We provide evidence for previously undescribed Archaea–host associations, including Bathyarchaeia and *Methanothermobacter*, the latter of which was prevalent among Aves and relatively abundant in species with higher body temperatures, although this association could not be decoupled from host phylogeny. Host phylogeny explained archaeal diversity more strongly than diet, while specific taxa were associated with both factors, and cophylogeny was significant and strongest for mammalian herbivores. Methanobacteria was the only class predicted to be present in the last common ancestors of mammals and all host species. Further analysis indicated that Archaea–Bacteria interactions have a limited effect on archaeal diversity. These findings expand our current understanding of Archaea–vertebrate associations.

Next-generation sequencing (NGS) has greatly expanded our view of archaeal diversity, which now consists of nearly 40 major clades, 8 of which are currently known to be host-associated[1,2]. Many of these clades consist of methanogens, which utilize bacterial fermentation products (namely hydrogen and carbon dioxide) for obtaining energy and are generally the most abundant Archaea in the mammalian gut[3,4]. Halobacteria, Thaumarcheota and Woesearchaeota comprise the major non-methanogenic host-associated archaeal clades and are generally not as prevalent or abundant among vertebrate gut microbiomes[2,5].

Most data on archaeal diversity in the vertebrate gut derives from studies using standard 'universall' 16S rRNA gene (16S) primers, which have recently been shown to grossly under-sample archaeal diversity relative to using Archaea-targeting 16S primers[6–8]. Therefore, much likely remains unknown of archaeal diversity and community assembly in the vertebrate gut. Setting primer issues aside, previous studies have identified host evolutionary history and diet to be the main factors influencing the gut microbiome[9–13]. Although some studies have shown specific evidence that gut archaeal diversity is dictated by host relatedness[14–18], focus has generally been on humans and certain mammalian clades. Still, diet may also play a significant role, especially given that fibre can increase methanogen levels and ruminants generate substantial amounts of methane[3]. Microbe–microbe interactions between Archaea and Bacteria may also have a strong influence on archaeal diversity, particularly syntrophic interactions between methanogens and bacterial fermenters[19–21]. Here, we characterize archaeal diversity in faecal/gut samples from 110 vertebrate species spanning five taxonomic classes, making this the largest reported Archaea-targeted study of vertebrate gut microbiome diversity. Using dietary and host phylogenetic relationships, as well as previously characterized bacterial diversity, we uncover robust relationships between Archaea, host phylogeny, and to some extent, host diet.

## Results

We utilized Archaea-targeting 16S primers that previously revealed vastly more gut archaeal community diversity in five great ape species relative to 'universal' 16S primers[6]. Our resulting gut microbiome 16S amplicon sequence data set consisted of 185 samples from 110 species comprising five vertebrate classes (Fig. 1, Supplementary Figs. 1 and 8 and Supplementary Tables 1 and 3). Most samples were derived from individual animals in the wild (76%), which is important given that captivity can alter the vertebrate gut microbiome[22,23]. Not all animal samples yielded adequate sequence data (Methods) to be included in the final data set (60% success; 185 of 311 samples; Supplementary Table 2). Failure was not correlated with host taxonomy, diet, other host characteristics, the amount of sample collected, the concentration or quality of genomic DNA (gDNA) or the Bacteria:Archaea ratio, as determined via metagenome sequencing (Supplementary Figs. 2–4). However, 16S rRNA gene copy number, as measured via quantitative polymerase chain reaction (qPCR) with 'universall' 16S primers, was significantly higher in the successful samples, suggesting that low microbial biomass was a major cause of failure (Supplementary Fig. 3b).

We found per-host archaeal diversity to be rather low, with only ~250 sequences saturating diversity estimates, regardless of host class or diet (Supplementary Fig. 7). Still, the taxonomic

[1]Department of Microbiome Science, Max Planck Institute for Developmental Biology, Tübingen, Germany. [2]TU Wien, Institute of Chemical, Environmental and Bioscience Engineering, Research Group for Environmental Microbiology and Molecular Diagnostics 166/5/3, Vienna, Austria. [3]ICC Interuniversity Cooperation Centre Water & Health, Vienna, Austria. [4]Wildlife Conservation Society, Bronx, NY, USA. [5]Research Institute of Wildlife Ecology, University of Veterinary Medicine, Vienna, Austria. [6]Research Division Water Quality and Health, Karl Landsteiner University for Health Sciences, Krems an der Donau, Austria. [7]Cluster of Excellence EXC 2124 Controlling Microbes to Fight Infections, University of Tübingen, Tübingen, Germany. ✉e-mail: nyoungblut@tuebingen.mpg.de

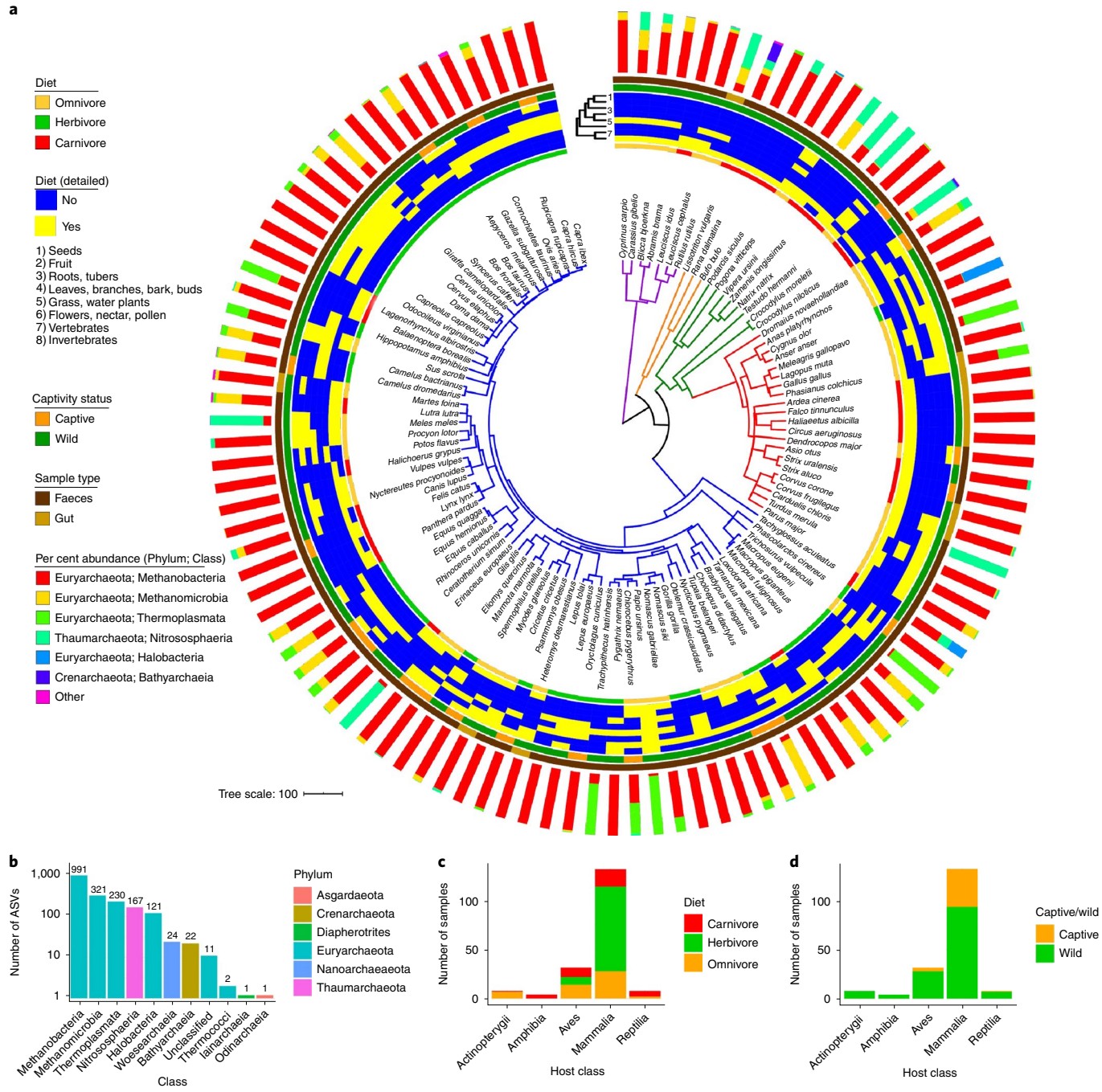

**Fig. 1 | Substantial prevalence and diversity of Archaea among vertebrates. a**, A dated phylogeny of all host species (*n* = 110) obtained from http://timetree.org, with branches coloured by host class (purple, Actinopterygii; orange, Amphibia; green, Reptilia; red, Aves; blue, Mammalia). For inner to outer, the data mapped onto the phylogeny are: host diet (general), detailed diet composition (the dendrogram depicts Jaccard similarity of dietary component presence/absence), wild/captive status, sample type and mean per cent abundances of archaeal taxonomic classes among all individuals of the species. **b**, Number of ASVs belonging to each class. **c,d**, Number of samples grouped by host class and diet (**c**) and host class and captive/wild status (**d**).

composition of the entire data set was rather diverse for Archaea, comprising six phyla and ten classes (Fig. 1). The data set consisted of 1,891 amplicon sequence variants (ASVs), with dramatic phylum- and class-level compositional variation among host species but relatively low variation within species (Supplementary Fig. 8 and Supplementary Table 4). Methanobacteria (Euryarchaeota phylum) dominated in the majority of hosts. In particular, a few of the 699 *Methanobrevibacter* ASVs were predominant, but they differed in abundance distributions across host clades and diets

(Supplementary Fig. 9). Thermoplasmata (Euryarchaeota phylum) dominated in multiple non-human primates, while two mammalian and one avian species were nearly completely comprised of *Nitrososphaeria* (Thaumarchaeota phylum): the European badger (*Meles meles*), the western European hedgehog (*Erinaceus europaeus*) and the rook (*Corvus frugilegus*). Halobacteria (Euryarchaeota phylum) dominated the goose (*Anser anser*) microbiome, which were all sampled from salt marshes. The class was also present in some distantly related animals (for example, the Nile crocodile

(*Crocodylus niloticus*) and the short beaked echidna (*Tachyglossus aculeatus*)) (Supplementary Table 4).

Of the ten observed archaeal classes, four are not known to include host-associated taxa[2]: Bathyarchaeia, Iainarchaeia, Odinarchaeia and Thermococci (Fig. 1). The most prevalent and abundant was Bathyarchaeia (Supplementary Fig. 6), which comprised nine ASVs present in six species from four vertebrate classes. It was rather abundant in the Nile crocodile (3.3%) and the two smooth newt samples (17.9% and 42.2%) (Supplementary Table 5). The other three classes comprised a total of four ASVs and were observed very sparsely and at low abundance, suggesting transience or persistence at very low abundances.

Only 40% of ASVs had a ≥97% sequence identity match to any cultured representative (Supplementary Fig. 10a). Of the ten archaeal taxonomic classes, five had no match at ≥85% sequence identity: Odinarchaeia, Bathyarchaeia, Iainarchaeia, Woesarchaeia and Thermococci. Taxonomic relatedness to cultured representatives differed substantially among the other five classes but was still rather low (Supplementary Fig. 10b), even for relatively well-studied clades (for example, Methanobacteria). These findings suggest that our data set consists of a great deal of uncultured taxonomic diversity.

Of 140 samples that overlap between our Archaea-targeted 16S data set (16S-arc) and that from our previous work with standard 'universal' 16S primers (16S-uni), 1,390 versus only 169 archaeal ASVs were observed in each respective data set (Supplementary Fig. 11). Representation of major clades was also much higher for the 16S-arc data set. For example, Methanobacteria was observed in all host species via the 16S-arc primers, while prevalence dropped substantially for 16S-uni primers (for example, only 9% for Aves).

We used multiple regression on matrices (MRM) to assess the factors that explain archaeal diversity. Notably, we employed a permutation procedure to assess the sensitivity of our results to archaeal compositional variation among hosts of the same species (Methods). Geographical distance, habitat and technical components (for example, faeces versus gut contents samples) did not significantly explain beta diversity, regardless of the diversity metric (Fig. 2a). Host phylogeny significantly explained diversity as measured by unweighted UniFrac, Bray–Curtis and Jaccard ($P < 0.05$); however, significance was not quite reached for weighted UniFrac. The per cent variation explained was dependent on the beta diversity measure and varied from ~28% for Jaccard to ~12% for unweighted UniFrac. In contrast to host phylogeny, composition of dietary components (diet) was only significant for Bray–Curtis, with ~12% of variance explained. Mapping the major factors onto ordinations qualitatively supported our results (Supplementary Fig. 12). Applying the same MRM analysis to just mammalian species maintained the strongest association with host phylogeny, although only Bray–Curtis and Jaccard distances were significant, possibly due to the lower sample sizes (Supplementary Fig. 13). MRM on just non-mammalian species did not generate any significant associations between host phylogeny or diet (Supplementary Fig. 14), probably due to the low sample sizes ($n = 39$). However, host phylogeny explained as much variance as including all species, whereas variance explained by diet was relatively small. Altogether, these findings suggest that host evolutionary history mediates vertebrate gut archaeal diversity more than diet.

We also assessed alpha diversity via MRM to provide a consistent comparison with our beta diversity assessment (Supplementary Fig. 15). No factors significantly explained alpha diversity calculated via either the Shannon Index or Faith's PD.

Although diet did not strongly explain total archaeal diversity, it may substantially explain the distribution of particular archaeal taxa. We used two methods to resolve the effects of diet on the archaeal microbiome while controlling for host evolutionary history: phylogenetic generalized least squares (PGLS) and randomization of

residuals in a permutation procedure (RRPP)[24,25]. RRPP and PGLS identified the same ten ASVs as being significantly associated with diet, while RRPP identified five more, probably due to increased sensitivity (adj. $P < 0.05$; Fig. 2b and Supplementary Fig. 16). All 15 ASVs belonged to the Euryarchaeota phylum and comprised four genera: *Methanobrevibacter*, *Methanosphaera*, *Methanothermobacter* and *Candidatus* Methanomethylophilus. RRPP model predictions of ASV abundances showed that Methanobacteria ASVs differed in their responses to diet, with five being most abundant in herbivores, whereas the other six were more abundant in omnivores/carnivores (Fig. 2b). Notably, diet enrichment differed even among ASVs belonging to the same genus. In contrast to the Methanobacteria ASVs, all four Methanomethylophilus ASVs were predicted as more abundant in omnivores/carnivores. These findings suggest that diet influences the abundances of particular ASVs, and even closely related ASVs can have contrasting associations to diet. All significant ASVs were methanogens, which may be due to the species studied (for example, a mammalian bias) or possibly because certain methanogens respond more readily to diet.

When applied to alpha or beta diversity, neither PGLS nor RRPP identified any significant associations with diet (adj. $P > 0.05$). These findings correspond with our MRM analyses by indicating that diet is not a strong modulator of total archaeal diversity.

We also assessed whether particular archaeal taxa are explained by host evolutionary history and identified 37 ASVs as having abundances correlated with host phylogenetic relatedness (Pagel's λ, adj. $P < 0.05$). These ASVs spanned three phyla: Euryarchaota, Thaumarchaeota and Crenarchaeota (Fig. 2c). The clade with the highest number of significant ASVs ($n = 15$) was Methanobacteriaceae, followed by Nitrososphaeraceae ($n = 12$) and Methanocorpusculaceae ($n = 5$). No such phylogenetic signal was observed when assessing alpha diversity rather than ASV abundances (adj. $P > 0.05$), which corresponds with our MRM results. We also tested for local instead of global correlations between ASV abundances and host phylogeny via the local indicator of phylogenetic association (LIPA). Twenty-five ASVs showed significant associations with certain host clades (Supplementary Figs. 17a and 18). For instance, three Nitrososphaeraceae ASVs were associated with two snake species (*Zamenis longissimus* and *Natrix natrix*), three *Methanobrevibacter* ASVs were associated with two species of kangaroo (*Macropus giganeus* and *Macropus fuliginosus*) and a *Methanocorpusculum* ASV was associated with both camel species (*Camelus dromedarius* and *Camelus bactrianus*). The two major exceptions to this trend were the *Methanothermobacter* ASVs, which associated with many species of Aves, whereas the *Methanobrevibacter* and *Methanosphaera* ASVs associated with many Artiodactyla species. Summarizing the number Archaea–host clade associations revealed clear partitioning of archaeal taxa by host clade, except for *Methanobrevibacter*, for which at least one ASV was associated with each of the host orders ($n = 23$ orders; Supplementary Fig. 17b).

To test for corresponding phylogenetic associations in both the host phylogeny and the archaeal 16S rRNA phylogeny, we employed two measures of cophylogeny: Procrustes Application to Cophylogenetic Analysis (PACo) and ParaFit[26,27]. Both PACo and ParaFit tests were significant ($P < 0.01$). We found significant differences in the distribution of PACo Procrustes residuals among the vertebrate classes and diets (Kruskal–Wallis <0.01; pairwise Wilcox <0.01 for all), indicating stronger signals of cophylogeny in the order Mammalia > Aves > Reptilia > Amphibia > Actinopterygii for taxonomy and herbivores > omnivores/carnivores for diet (Fig. 2d,e).

We utilized ancestral state reconstruction (ASR) to investigate which archaeal clades were probably present in the ancestral vertebrate gut. Predictions of class-level abundances were accurate for extant hosts (adj. $R^2 = 0.86$, $P < 2 \times 10^{-16}$; Supplementary Fig. 19), and all 95% confidence intervals (CIs) were constrained

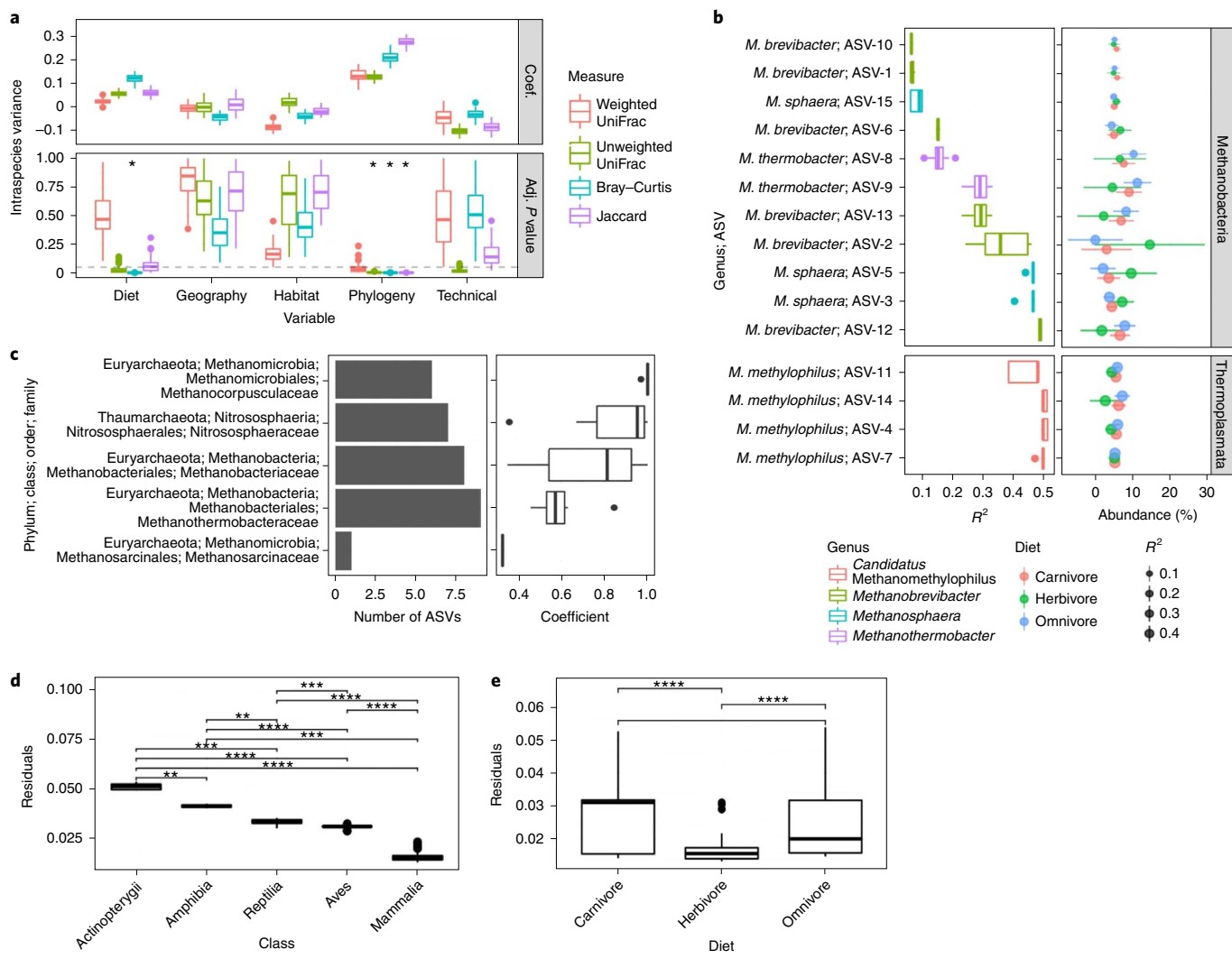

**Fig. 2 | Host phylogeny and diet significantly explain different aspects of archaeal diversity. a**, The distribution of partial regression coefficients (Coef.) and $P$ values (Adj. $P$ value) across 100 data set permutations used for MRM tests (two-sided). For each permutation, one individual per host species was randomly sampled. MRM tests assessed the beta diversity variance explained by host diet, geography, habitat, phylogeny and 'technical' parameters (Methods). The box plots describe variance in effect sizes observed among each data set permutation ($n = 100$). Asterisks denote significance (adj. $P < 0.05$ for >95% of data set subsets; see Methods). **b**, ASVs in which abundances are significantly correlated with diet (adj. $P < 0.05$) while controlling for host phylogeny via RRPP. The left plot shows the distribution of coefficient values across all 100 permutations of the host tree, while the right plot shows RRPP model predictions of ASV abundances, depending on diet (points = mean; line ranges = 95% CI). **c**, The left plot shows the number of ASVs with a significant global phylogenetic signal (Pagel's $\lambda$, adj. $P < 0.05$), while the right plot shows the distribution of coefficient values for those ASVs. **d,e**, The distribution of PACo residuals across samples (averaged across all 100 data set permutations) and grouped by host class (**d**) or diet (**e**). Brackets with asterisks indicate significant pairwise differences (Wilcox two-sided, **adj. $P < 0.01$, ***adj. $P < 0.001$, ****adj. $P < 0.0001$). Box centre lines, edges, whiskers and points signify the median, interquartile range (IQR), 1.5× IQR and >1.5× IQR, respectively. See the statistical source data for all other statistical information.

enough to be informative (26% ± 29 s.d.). The model revealed that Methanobacteria was uniquely pervasive across ancestral nodes, whereas other classes were distributed sparsely among extant taxa and across a few, more recent ancestral nodes (Fig. 3 and Supplementary Fig. 20). Importantly, the model predicted that Methanobacteria was the only class to be present in the last common ancestor (LCA) of all mammals and the LCA of all five host taxonomic classes (Fig. 3f,g). We generated a similar ASR model for all four Methanobacteria genera. Our model was more accurate at predicting extant traits than our class-level model (adj. $R^2 = 0.93$, $P < 2 \times 10^{-16}$; Supplementary Fig. 19) and 95% CIs were informative (28% ± 24 s.d.). The model predicted three of the four genera to be present in the LCA of all mammals and the LCA of all host

species (Fig. 3f,g). *Methanobrevibacter* and *Methanothermobacter* were predicted to have similar abundances for both LCAs (~30–35%), whereas *Methanosphaera* was much lower (~5%). The model predicted *Methanobrevibacter* to be most highly abundant in the Artiodactyla and generally abundant across most mammalian clades (Supplementary Fig. 21), whereas *Methanothermobacter* was predicted to be most highly abundant and prevalent across the Aves and also mammalian clades in which *Methanobrevibacter* was less abundant (for example, Carnivora and Rodentia). *Methanosphaera* was prevalent across most animal clades, but generally at low abundance. Importantly, we found both ASR analyses (class and genus levels) to be robust to biases in the number of samples per host species (Supplementary Fig. 22).

*Methanothermobacter* is not known to be host-associated[2]. However, a total of 39 *Methanothermobacter* ASVs were observed across 78 samples ($18 \pm 30$ s.d. samples per ASV), which strongly suggests that its presence is not due to contamination. Moreover, the top BLASTn hit for 36 of the 39 ASVs was to a cultured *Methanothermobacter* strain (Supplementary Fig. 23 and Supplementary Table 6), which indicates that the taxonomic annotations are demonstrably correct. The high prevalence of *Methanothermobacter* among Aves led us to the hypothesis that body temperature significantly affects the distribution *Methanothermobacter* (Supplementary Fig. 24), given that birds generally have higher body temperatures than mammals[28] and all existing *Methanothermobacter* strains are thermophiles[29]. Moreover, *Methanothermobacter* is not abundant in Monotremata and Marsupialia species relative to the placental groups, which reflects the higher body temperature of placentals (Supplementary Fig. 24). We were able to assign published body temperature data to 73 mammalian and avian species (Supplementary Fig. 25a,b and Supplementary Table 7). Genus-level abundances of *Methanothermobacter* significantly correlated with body temperature (RRPP, adj. $P < 0.001$), whereas *Methanobrevibacter* and *Methanosphaera* did not (Supplementary Fig. 25c,d). However, the association was only significant if not accounting for host phylogeny (RRPP, adj. $P > 0.05$), indicating that the association between *Methanothermobacter* and body temperature could not be decoupled from host evolutionary history. We also identified seven *Methanothermobacter* ASVs to be correlated with body temperature (RRPP, adj. $P < 0.05$; Supplementary Fig. 25e), whereas no *Methanobrevibacter* or *Methanosphaera* ASVs were correlated. Again, the association was only significant if not accounting for host phylogeny. Regardless, we provide evidence congruent with the hypothesis that *Methanothermobacter* abundance is modulated by host body temperatures (Supplementary Figs. 24 and 25). We note that among the host species in which methane emission data exist[14,30], avian species with high abundances of *Methanothermobacter* have emission rates at the higher end of mammal emission rates (Supplementary Fig. 29), suggesting that *Methanothermobacter* is indeed a persistent inhabitant in the gut of some avian species.

qPCR with newly created *Methanothermobacter*-targeting primers partially supported our findings that *Methanothermobacter* is present among many avian and mammalian species (Supplementary Fig. 27). Notably, *Methanothermobacter* presence/absence was consistent for members of the same host species. *Methanothermobacter* gene copies per gram of gastrointestinal (GI) sample varied from ~$5 \times 10^3$ to $5 \times 10^6$. Although *Methanothermobacter* was more sparsely observed among Aves via qPCR versus in the ASV data, this may have resulted from primer biases or lack of enough high quality gDNA for qPCR.

Metagenome assembly of avian samples also supported our findings of high *Methanothermobacter* prevalence across Aves. We mapped all assembled contigs ($\geq 1.5$ kb) to species-representative genomes of the Methanothermobacteriales (GTDB Release 95 taxonomy). The number of contigs that mapped to Methanothermobacteraceae or Methanothermobacteraceae A was $88 \pm 799$ (s.d.), which was $0.88\% \pm 0.93$ (s.d.) of all contigs assembled (Supplementary Fig. 28). However, we were not able to assemble a genome classified as the target clade, possibly due to high intrasample strain diversity, as suggested by the ASV data (Supplementary Table 6).

In addition to host-specific factors modulating diversity, microbe–microbe interactions may also play a significant role. A co-occurrence network of all archaeal ASVs revealed high assortativity by taxonomic group, regardless of the taxonomic level (Supplementary Figs. 30 and 31). The only significant negative co-occurrences were between a cohort dominated by *Methanobrevibacter* and one dominated by *Methanothermobacter*. These two cohorts differed substantially in their distributions across host clades, with the *Methanobrevibacter*-dominated cohort highly prevalent only among Artiodactyla, whereas the *Methanothermobacter*-dominated cohort was prevalent across a number of mammalian orders (for example, Carnivora and Rodentia) and almost all avian orders (Supplementary Fig. 32). We assessed whether taxonomic assortativity differs among host diets (Supplementary Fig. 33), and found assortativity to be lowest for omnivores and highest for carnivores, suggesting that the carnivore gut is composed of simpler and more taxonomically homogenous archaeal consortia relative to omnivores and herbivores.

We assessed Archaea–Bacteria interactions by comparing the overlapping 16S-arc and 16S-uni samples ($n = 140$). Archaeal and bacterial alpha and beta diversity were not correlated, regardless of measure (adj. $P > 0.05$; Fig. 4a,b), which suggests that total archaeal diversity is not explained by bacterial diversity or vice versa. Overall network taxonomic assortativity was low; however, the assortativity of just Archaea was quite high ($\geq 0.774$ for all taxonomic levels; Fig. 4e). The most common Archaea–Bacteria associations were between an unclassified Methanomicrobia genus and *Citrobacter*, along with an unclassified Nitrososphaeraceae genus and *Enterococcus* (Supplementary Table 9). As with the Archaea-only network, there were two cohorts dominated by different Archaea: the first by *Methanobrevibacter* and the second by *Methanothermobacter* (Fig. 4c,d). These two cohorts differed dramatically in bacterial diversity: the *Methanobrevibacter* cohort comprised 13 bacterial families from three phyla, whereas the *Methanothermobacter* cohort included only three families from two phyla (Supplementary Fig. 34).

## Discussion

We show that the vertebrate gut harbours a great deal more archaeal diversity than previously observed (Fig. 1 and Supplementary Fig. 11)[13,22]. We were unsuccessful at obtaining sufficient archaeal amplicon sequence data from many samples (40% failure rate), which appears to be a result of low microbial biomass in some samples (Supplementary Figs. 2–4). However, the broad host taxonomic diversity of our final data set suggests that Archaea are widespread among vertebrates (Fig. 1 and Supplementary Fig. 4). Moreover, the diversity of archaeal ASVs across samples and repeated intrahost species observations of archaeal taxa indicate that the vertebrate gut microbiome collectively harbours a diverse archaeal assemblage.

Little is known of the gut microbiome for many of the host species in our data set, especially regarding archaeal diversity. Only a minority of all known archaeal phyla and classes include any cultured representative[1], and accordingly we show that the majority of ASVs in our data set lack cultured representatives (Supplementary Figs. 10 and 23), even ASVs belonging to more well-studied archaeal clades (for example, *Methanobrevibacter*). Interestingly, our data set included repeated observations of archaeal clades not known to inhabit the vertebrate gut[2], suggesting previously unknown Archaea–vertebrate associations. For instance, the Bathyarchaeia

**Fig. 3 | Ancestral state reconstruction evidences Methanobacteria association with ancestral vertebrate gut. a**, Predicted abundances of Methanobacteria for each extant host species (yellow circles) and ancestral host species (blue circles). Circle size denotes relative abundance (min = 1%, max = 100%). The phylogeny is the same as shown in Fig. 1. **b,c**, Estimated Methanobacteria abundance (points) with 95% CIs (line ranges) for the LCA of all mammalian species (**b**) and all five taxonomic classes (**c**). **d,e**, The trees are as in **a** but show the predicted abundances of the *Methanobrevibacter* and *Methanothermobacter* genera, respectively. **f,g** Plots as in **b** and **c**, respectively, but showing predictions of abundances for all four genera in the Methanobacteria class. The sample sizes for **b**, **c**, **f** and **g** are all 110. See the statistical source data for all other statistical information.

(also known as *Candidatus* Bathyarchaeota) is not known to include host-associated members[2], but this clade consisted of nine ASVs and was substantially abundant in multiple individuals

comprising various, distantly related species (Fig. 1). Bathyarchaeia currently lacks any cultured representatives[1], but inference from metagenome-assembled genomes suggests that the clade contains

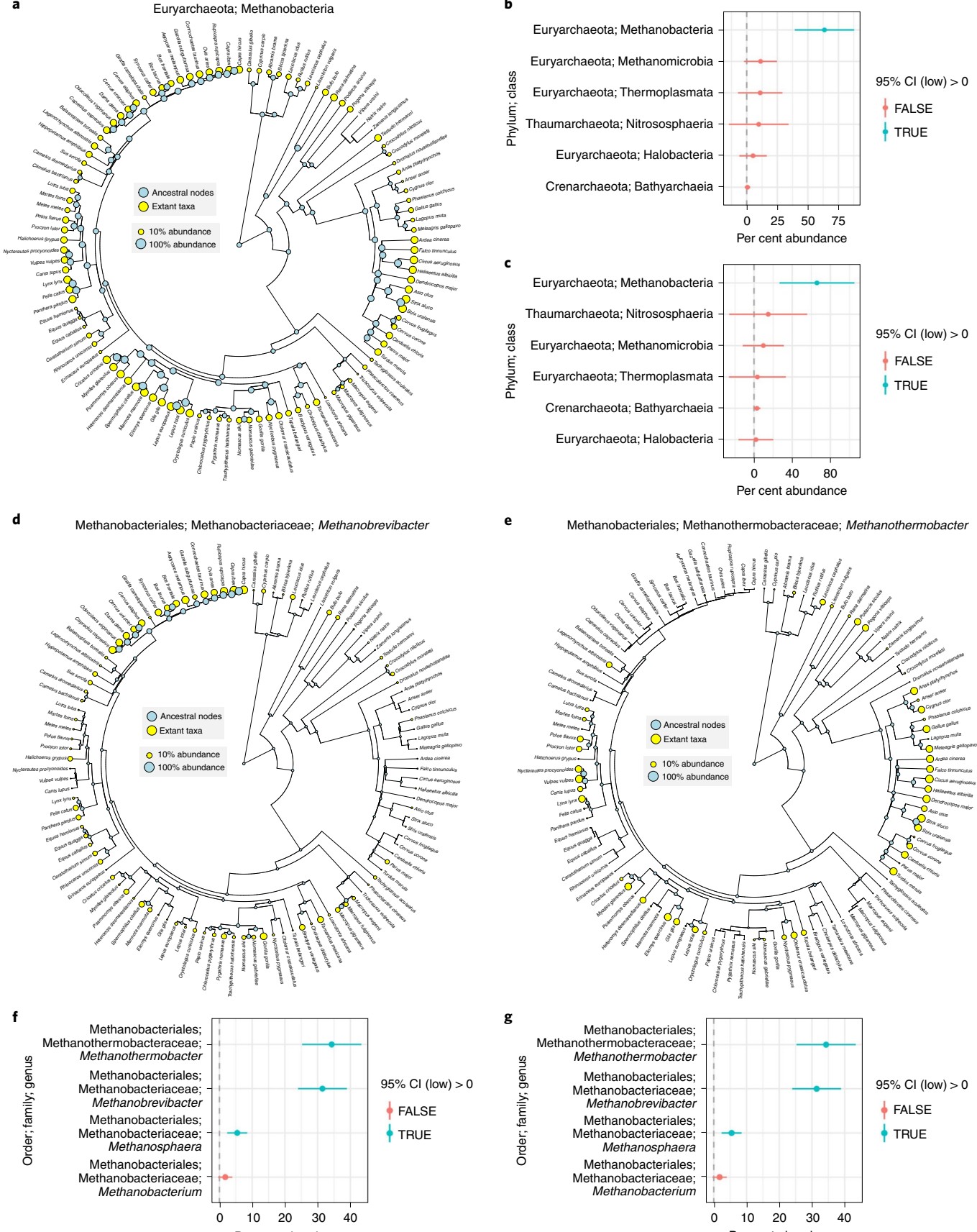

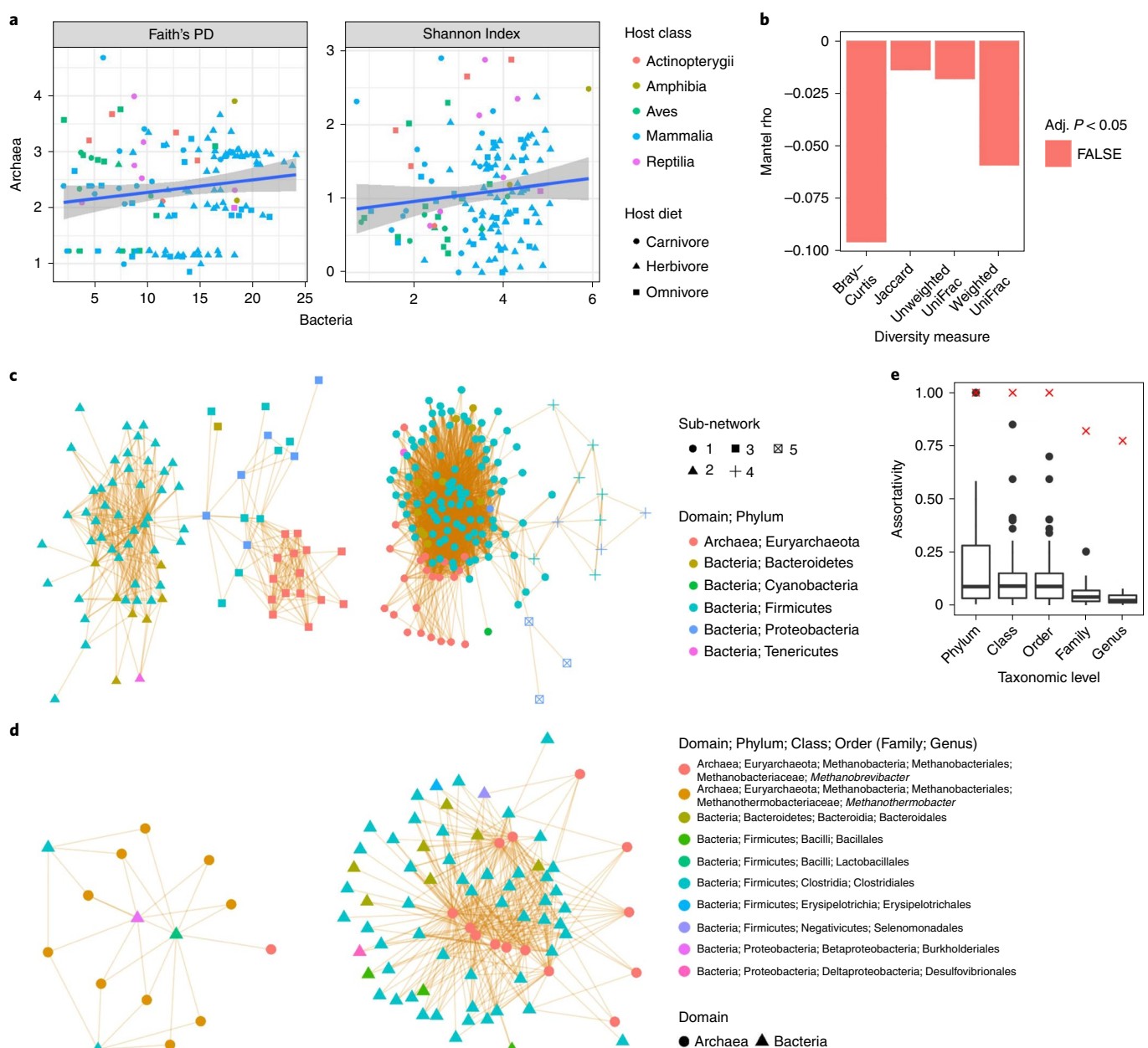

**Fig. 4 | Limited associations between archaeal and bacterial diversity. a**, Linear regressions (blue lines) of archaeal versus bacterial community diversity as measured via Faith's PD and the Shannon Index. The grey areas denote 95% CIs. **b**, Mantel tests comparing archaeal and bacterial beta diversity (999 permutations per test). **c**, Archaea–Bacteria ASV co-occurrence network, with nodes coloured at the phylum level. Orange edges indicate significant positive co-occurrences. **d**, Subnetworks only (determined via the walktrap algorithm; see Methods) of the co-occurrence network shown in **c** that contain archaeal ASVs. **e**, Taxonomic assortativity of just Archaea–Archaea edges (red X) versus just Bacteria–Bacteria edges (box plots), with Bacteria subsampled to the same number of ASVs as Archaea (100 subsample permutations). Box centre lines, edges, whiskers and points signify the median, IQR, 1.5× IQR and >1.5× IQR, respectively. See the statistical source data for all other statistical information.

methanogens and homoacetogens, and has been detected across a wide range of environments, especially aquatic biomes[31,32]. The relatively high abundance of Bathyarchaeia in the smooth newt and the Nile crocodile, both of which have semi-aquatic lifestyles, may suggest adaptation to the gut from sediment-inhabiting ancestors. Alternatively, consumption of sediment may result in its transitory presence in the vertebrate gut, but one would then expect more Bathyarchaeia presence in other (semi-)aquatic vertebrates ($n = 24$) versus what we observed (Fig. 1; Supplementary Table 5).

*Methanothermobacter* is another clade not known to be host-associated, especially given the high optimal growth temperature

(55–66 °C) of existing cultures[29,33]. Regardless, we observed a diverse set of *Methanothermobacter* ASVs that were most prevalent and abundant among avian species (Supplementary Fig. 24). Moreover, qPCR with *Methanothermobacter*-targeting primers and metagenome assembly partially supported the ASV data by providing multiple lines of evidence that *Methanothermobacter* is present in many avian species (Supplementary Figs. 26–28). How is *Methanothermobacter* inhabiting the relatively cool gut environment, and why is this clade not considered host-associated? First, some *Methanothermobacter* isolates can grow at 37 °C, but not optimally[33]. Second, *Methanothermobacter* has been observed in a

number of animals including salamanders[34], trout[35], chickens[36] and buffalo[37], but very sparsely and at low abundances. Third, we found *Methanothermobacter* abundances to correlate with higher body temperatures of avian and some mammalian species which have generally been less-studied, especially in regards to archaeal diversity (Supplementary Figs. 24 and 25).

We identified factors that significantly explain interspecies variation in archaeal community composition. Overall, habitat did not significantly explain interspecies beta diversity (Fig. 2a); however, the high abundance of halophilic archaea in host species inhabiting high-saline habitats (for example, geese) hints at habitat playing a role in certain cases (Fig. 1). Both host phylogeny and diet correlated with beta diversity, but host phylogeny was more consistently significant among diversity metrics and showed stronger correlations (Fig. 2a). These findings suggest that host evolutionary history has a stronger effect on total archaeal community diversity among host species relative to diet (both general dietary categories and specific components), which corresponds with existing evidence of an association between archaeal diversity and host phylogeny[12,14]. However, these results contrast with our previous Bacteria-biased survey of the same sample data set, in which diet was an equal if not stronger explanatory variable relative to host phylogeny[13].

This difference between how diet associates with bacterial and archaeal diversity may be a result of three major factors. First, methanogens dominate most archaeal communities in our data set (Fig. 1), and these Archaea may not respond readily to diet because they only secondarily consume a relatively homogenous set of bacterial fermentation products[2,4]. Second, the nitrifying Archaea that dominate certain archaeal communities may be strongly influenced by the nitrogenous content of dietary components, which we did not directly measure. For instance, very high abundances of Thaumarchaeota in certain vertebrates (that is, European badger, western European hedgehog and rook) may be the result of consuming invertebrates such as earthworms, pill bugs and termites; all of which all have been observed to harbour gut-inhabiting Thaumarchaeota[38-40]. Third, diet may only strongly influence a minority of archaeal ASVs, given that we only observed significant associations between ASV abundances and host diet for 15 ASVs (Fig. 2b and Supplementary Fig. 16). Interestingly, closely related Methanobacteria ASVs showed contrasting associations with diet, suggesting niche partitioning at fine taxonomic scales, possibly due to differing syntrophic associations with bacterial fermenters. Indeed, the pan-genome of *Methanobrevibacter smithii* is highly variable in adhesin-like proteins thought to mediate cell–cell interactions with specific bacterial syntrophs[16,41].

Although previous work has indicated that host phylogeny influences archaeal abundance[12-14,30,42], it focused largely on methanogen diversity (especially *Methanobrevibacter*) and was strongly biased towards mammals. In this study, we not only identified a signal of host–Archaea phylosymbiosis while accounting for host diet, habitat and other factors (Fig. 2a), but we also identified specific archaeal ASVs to be associated with host phylogeny globally and for particular host clades (Fig. 2c and Supplementary Fig. 17), and we observed a significant signal of cophylogeny (Fig. 2d,e). Although both the signal of phylosymbiosis and cophylogeny may be dominated by methanogens, we did identify specific non-methanogenic ASVs to be associated with host phylogeny (Supplementary Fig. 17), indicating that the influence of host phylogeny is not restricted to methanogens. These non-methanogenic ASVs were all Thaumarchaeota and showed host clade specificity to reptiles and bony fish (Supplementary Fig. 17), which emphasizes a need to study non-methanogenic Archaea outside mammals and birds. Interestingly, all significant local phylogenetic signals for mammals were herbivorous species. These results concur with our assessments of cophylogeny, which show the strongest signal for mammals and herbivores (Fig. 2d,e). We previously observed a similar

result when focusing on the bacterial community[13], suggesting that both host–Archaea and host–Bacteria cophylogeny is strongest for herbivorous mammals. Many characteristics typical of mammals may be driving this pattern, such as complex gut morphology and placental birth[43,44].

Based on methane emission data, Hackstein and van Alen hypothesized that methanogens were present in the gut of the Mammalia LCA and probably even present in the Vertebrata LCA, but certain lineages have permanently lost this 'traitl'[14,42]. Our ancestral state reconstruction of archaeal abundances did indeed support methanogen presence in the LCA of mammals and all five taxonomic classes of vertebrates (Fig. 3), but importantly, we specifically identified Methanobacteria as the only archaeal class to show such evidence. Moreover, we evidenced three of the four Methanobacteria genera (*Methanobrevibacter*, *Methanothermobacter* and *Methanosphaera*) to be present in the LCA of mammals and all five classes (Fig. 3). The prevalence of these three genera across all vertebrate classes suggests that methanogens are not completely lost from certain lineages but rather just differ in relative abundance (Fig. 3). The relatively high predicted abundance of *Methanothermobacter* in the LCA of all host species may be biased due to its high abundance in most avian species and the limited number of ectothermic host species. Nevertheless, our findings suggest a long evolutionary association between vertebrates and certain methanogens.

We found little evidence of microbe–microbe interactions influencing archaeal diversity. Archaea generally assorted with sister taxa, regardless of diet (Supplementary Figs. 30 and 33), which may be the result of strong selective pressure counteracting competitive exclusion. Indeed, we observed niche partitioning by diet among closely related *Methanobrevibacter* ASVs (Fig. 2b, Supplementary Figs. 9 and 16). Moreover, closely related *Archaea* may be spatially partitioned along the GI tract as observed in human biopsies[7], or along other niche axes such as virus–Archaea interactions[45]. Archaeal taxonomic assortativity was high relative to Bacteria (Fig. 4e), indicating that community assembly processes differ between the domains, which corresponds with the lack of correlation between archaeal and bacterial diversity (Fig. 4a,b). These findings indicate that methanogen–Bacteria syntrophic interactions do not strongly dictate archaeal diversity[2,4,46]. Interestingly, we observed two major Archaea–Bacteria assemblages dominated by *Methanobrevibacter* and *Methanothermobacter*, respectively. These assemblages reflect the different distributions of each methanogen genus, which may be influenced by body temperature (Supplementary Figs. 24 and 25). Given that members of both methanogen clades are hydrogenotrophs[29], possess adhesins for attachment to organic surfaces[41,47] and share many homologues via horizontal gene transfer and shared common ancestry[37], body temperature may be a main niche axis partitioning these two clades.

In conclusion, our findings expand the paradigm of Archaea–host interactions, including previously undescribed host–Archaea associations for clades previously considered to not be host-associated (for example, *Methanothermobacter*). Although our methods allowed us to mitigate the variable number of samples per species (Methods), the differing number of samples per host clade (for example, the mammalian bias) may have influenced our findings; however, we note that many of our analyses assessed patterns at the level of host taxonomic class (for example, Fig. 2d, Supplementary Figs. 13 and 17). The broad taxonomic scope of our analyses allowed us to leverage interspecies evolutionary and ecological relatedness to generate robust findings, despite the limited number of samples for some host species. More work is needed to elucidate the role of host body temperature in modulating archaeal community assembly in the vertebrate gut, especially in regards to *Methanothermobacter* and *Methanobrevibacter*. Moreover, little is known of the potential host and microbial genetic factors underlying our observed signals of phylosymbiosis and cophylogeny.

One promising approach is to combine genome-capture for obtaining genome assemblies of rare microbial taxa[48] with microbiome genome-wide association analyses focused on archaeal strains[49].

## Methods

**Sample collection.** Collection of GI samples (that is, faeces or gut contents) was as described by Youngblut and colleagues[13]. Samples used in this study were collected between February 2009 and March 2014. Supplementary Table 1 lists all dates, locations and other relevant metadata associated with each sample. Diet and taxonomy metadata was compiled from the NCBI Taxonomy browser and the PanTHERIA database[50]. Samples were collected primarily by wildlife biologists conducting long-term research on the respective species in its habitat, which ensured that sampling guidelines and restrictions were adhered to, where applicable. Only fresh samples with a confirmed origin from a known species were collected. All samples were collected in sterile sampling vials, transported to a laboratory, and frozen within 8 h. DNA was extracted with the PowerSoil DNA Isolation Kit (MoBio Laboratories). We included negative controls (that is, PCR-grade water) in all batches of DNA extractions and measured each negative control with bacterial 16S rRNA gene targeted qPCR (see below for qPCR methodology). Extraction negative controls were always below the limit of detection for the qPCR assay.

**16S rRNA gene sequencing and data processing.** PCR amplicons for the V4 region of the 16S rRNA gene were generated with primers arch516F–arch915R[6,51] and were sequenced with the Illumina MiSeq 2 × 250 v.2 kit at the Max Planck Institute for Developmental Biology. DADA2 v.1.10.0 (ref. [52]) was used to generate ASVs. Taxonomy was assigned to ASVs with the QIIME2 q2-feature-classifier v.2019.10.0 (ref. [53]) using the SILVA database (v132)[54]. All ASVs not classified as Archaea were removed.

Although none of our negative controls produced substantial PCR products, we sequenced each. Valid 16S rRNA sequence data was obtained from each negative control, but the total number of sequences were very low (Supplementary Fig. 5) relative to any other sample, suggesting that the biomass of any contamination was orders of magnitude lower than the microbiome gDNA used for PCR and sequencing. Moreover, all negative controls were dominated by six Gram-positive bacterial genera: *Catellicoccus* (Firmicutes), *Catenibacterium* (Erysipelotrichia), *Lactobacillus* (Bacilli), *Bacillus* (Bacilli), *Lactococcus* (Firmicutes) and *Ruminococcaceae UCG-014* (Firmicutes).

Rarefaction analysis using alpha diversity quantified via the Vegan v.2.5-6R package (Shannon Index[55]) or the iNEXT v.2.0.19R package (Hill numbers: order = 1)[56] revealed that archaeal diversity saturated at a sampling depth of ~250 (Supplementary Fig. 7). Therefore, the data set was rarefied to this depth, with all samples lacking this depth filtered out. Because of the low prevalence of ASVs across host species (1.8% ± 23 s.d.), we did not employ the standard compositional data analysis transformation of centred log ratio, given the large number of zero values in the data set that would need to be imputed as non-zero values before the transformation (Supplementary Fig. 6). We found such imputation by either using a pseudo count of 1 or imputing via the Bayesian-multiplicative replacement method implemented in the zCompositions v.1.3.3R package[57] generated unrealistic distributions. QIIME2 was used to calculate alpha and beta diversity. To limit saturation of star phylogeny beta diversity measures (that is, no overlap of any ASVs across samples leading to maximum diversity values), we first aggregated ASV counts at the genus level. A phylogeny was inferred for all ASV sequences with fasttree[58] based on a multiple sequence alignment generated by MAFFT v.7.310 (ref. [59]). All samples lacking relevant metadata used in the study were filtered from the data set. In cases in which an individual host was sampled multiple times, we randomly selected one sample.

Samples from the 16S rRNA gene amplicon data set of Youngblut and colleagues were previously sequenced and processed in the same manner as done for the arch516F–arch915R amplicon data set, with the exception that primers 515F–806R were used and samples were rarefied to a depth of 5,000 (ref. [13]). To compare ASVs classified as Archaea in each data set, we filtered out all non-archaeal ASVs. For our analyses of Bacteria–Archaea interactions, we removed all archaeal ASVs from the 515F–806R data set. Alpha and beta diversity were calculated as stated above on genus-level relative abundances.

**Host phylogeny.** Only 21% of animals in our data set have existing genome assemblies of any quality in which to infer a genome-based phylogeny from. Instead, we used a dated host phylogeny for all species from http://timetree.org[60]. We created a phylogeny for all samples by grafting sample-level tips into each species node with a negligible branch length (Supplementary Fig. 8).

**Intraspecies sensitivity analysis.** The data set consisted of a differing number of samples per host species and no intraspecies phylogenetic relatedness data. Instead of just randomly subsampling one sample per species or using branches of zero length for phylogeny-based hypothesis testing, we employed a sensitivity analysis to assess robustness to intraspecies variability. The sensitivity analysis was performed as described by Youngblut and colleagues[13]. Briefly, for each hypothesis

test, we generated 100 permutation data sets in which one sample was randomly selected per species. A hypothesis test was considered robustly significant if >95% of the permutation data sets generated a significant result ($P < 0.05$, unless otherwise noted).

**qPCR with universal 16S rRNA gene primers.** qPCR was performed as described by Savio and colleagues[61]. Briefly, we used the 16S rRNA gene-targeting primers 8F and 338R[62], which specifically target the V1–V2 hypervariable regions. PCR reactions contained 2.5 µl of 1:4 and 1:16 diluted DNA extract as the template, 0.2 µM of each primer and iQ SYBR Green Supermix (Bio-Rad). Supplementary Table 10 lists gene copy number estimates.

**Design of qPCR primers targeting *Methanothermobacter*.** We designed CoreGenomePrimers (https://github.com/leylabmpi/CoreGenomePrimers), which is software to generate clade-specific primers that target a core gene in the pan-genome of the focal clade. The general workflow of the software is shown in Supplementary Fig. 26a. The input is a set of genomes for the target taxonomic clade (for example, *Methanothermobacter*), which can be obtained from Genbank RefSeq, or other sources. Prokka is used to call genes for each genome. All amino acid gene sequences for each genome are clustered via the MMseqs2 'clusterl' algorithm[63], but we note that the 'linclustl' algorithm can be used for large genome data sets. MAFFT[59] is used to generate an amino acid multiple sequence alignment for each core gene cluster, which is defined by default as single copy genes found in all members of the target clade. The amino acid alignments are reverse translated via revTrans[64].

CoreGenomePrimers utilizes Primer3 (ref. [65]) to generate candidate primer sets, given its ability to rapidly generate large numbers of primers based on user-provided parameter constraints such as minimum, maximum and optimal melting temperature or amplicon size. However, Primer3 cannot generate degenerate primers for an alignment but instead only generates primers for a single target sequence. We therefore utilized the majority consensus sequence of each alignment for primer design via Primer3. Each generated primer set was then mapped to the alignment, and degeneracies were added to each primer sequence to encompass all sequence variation at the target positions in the alignment. Primer sets were then filtered based on total degeneracies and degeneracies at the 3′ region (by default defined as the final five nucleotides).

Candidate primers are filtered based on off-target hits, first to other genes in the target genomes and then to any non-target clade. BLASTn-short is used for mapping primers during both filtering steps, with hits filtered if the alignment length is more than 80% of the primer length (that is, excluded if insufficient coverage to be considered a viable hit). Primer sets are filtered if both the forward and reverse primers hit on the correct strand within an acceptable distance to produce a viable amplicon. By default, primer sets producing non-target amplicons of 30–2,000 bp are filtered. A final table of primer sets lists the primer sequences, the gene cluster targeted and parameters about each primer such as average melting temperature (averaged across all non-degenerate sequence combinations for the degenerate primers).

In addition to the final table of primer sets, information is provided in regards to the annotations of each core gene cluster and the closest related gene sequences outside of the target clade. Gene annotations are provided via Prokka, whereas BLASTx versus NCBI number is used to identify the most closely related non-target gene sequences.

We note that CoreGenomePrimers can be used to generate qPCR primer sets that include an internal probe, which is designed by Primer3. Also, CoreGenomePrimers can be used to design primers for other PCR applications such as clade-targeted amplicon NGS.

To design *Methanothermobacter*-specific primers via CoreGenomePrimers, we obtained all available *Methanothermobacter* genomes from Genbank. We filtered genomes with CheckM-estimated completeness and contamination scores of <90% and >5%, respectively. We also de-replicated genomes via dRep at 99.9% average nucleotide identity (ANI) to filter any very closely related genomes. The remaining 16 *Methanothermobacter* genomes were used for primer design via CoreGenomePrimers (Supplementary Table 11). For gene clustering via the MMseqs2 'clusterl' algorithm, we used an 80% sequence identity and 90% coverage cut-off, respectively.

The most promising candidate primer set generated by CoreGenomePrimers (174FR) targeted a gene cluster annotated as a 30S ribosomal protein. The forward and reverse primer sequences are 5′-TMARRACMCACTGCAGGGAC-3′ and 5′-TCCGTGYTCAACYTTYTTCCT-3′, respectively. These primers were selected based on their low overall degeneracy, lack of any 3′ degeneracies (defined as the last five nucleotides), and similar melting temperatures of 60.4 and 59.5 °C, respectively. Moreover, the targeted gene cluster did not have any close BLAST hits to anything in NCBI nr, with 81% sequence identity for the closest non-target clade BLAST hit (MBC7101283.1, hypothetical protein).

We tested the 174FR primer set for clade-specific amplification by PCR screening the primers on a gDNA panel of methanogen and bacterial isolates (Supplementary Fig. 26b). Each PCR consisted of 25 µl of DreamTaq Green MasterMix (2×; ThermoFisher Scientific), 2.5 µl each of the forward and reverse primers (10 µM; Integrated DNA Technologies), 1 µl of gDNA (2 ng µl⁻¹) and 19 µl

of PCR-grade water, for a total PCR volume of 50 μl. The thermocycler conditions consisted of 1 min at 95 °C, followed by 30 cycles of 15 s at 95 °C, 15 s at either 61 or 62 °C, and 30 s at 72 °C, with a final extension for 5 min at 72 °C. The 174FR primer set only generated amplicons for the *Methanothermobacter* isolates (Supplementary Fig. 26b), which was expected based on our BLAST results showing that no non-target gene sequence was closely related to the target gene (above).

**qPCR with *Methanothermobacter*-targeting primers.** qPCR was carried out on various avian and mammalian GI gDNA samples with the *Methanothermobacter*-targeting primers 174FR (see above for information on primer design). *Methanothermobacter marburgensis DSM-2133* gDNA was used as a standard, with a $\log_{10}$-fold dilution curve from $5.31 \times 10^7$ to $5.31 \times 10$ gene copies per μl. Primer efficiency testing revealed that 400 nM of each primer produced a high PCR efficiency of 99.3% and a high standard curve $R^2$ of 0.982. *Methanothermobacter thermautotrophicus DSM-1053* gDNA was used as a positive control. *Roseburia hominis DSM-16839* and PCR-grade water were both used as negative controls, and neither generated critical threshold values. For each of three replicate PCRs, 5 ng of GI sample of gDNA was used, when possible. Samples were selected based on availability of enough gDNA (>5 ng in total). Each PCR consisted of 10 μl of KiCqStart SYBR Green qPCR ReadyMix (2×; Sigma-Aldrich), 0.8 μl of each primer (10 μM; Integrated DNA Technologies), 5.4 μl of PCR-grade water and 3 μl of gDNA, for a total volume of 20 μl. The thermocycler conditions consisted of 1 min at 95 °C, followed by 40 cycles of 15 s at 95 °C, 15 s 61 °C, 30 s at 72 °C, with a final extension for 30 s at 72 °C and then a melt curve analysis (55–95 °C, 5 s per set, 0.5 °C increase per step). The melt curve analysis did not show any indication of non-target amplicons for any PCR. Moreover, the products of select PCRs were visualized via 2% agarose gels to verify correct amplicon sizes. A Bio-Rad CFX Connect Real Time System was used for all qPCR experiments. Gene copies per gram of sample was calculated based on the mass of GI material used for gDNA extraction.

**Metagenomics.** We utilized existing metagenome sequence data from BioProject PRJEB38078 (ref. [66]) and also more deeply sequenced certain avian samples (Supplementary Fig. 28). Metagenome sequencing and data processing was done as in Youngblut and colleagues[66]. Briefly, NGS libraries were prepared as in Karasov and colleagues[67], and pooled libraries were sequenced via an Illumina HiSeq3000 instrument with 2 × 150 paired-end sequencing. The sequence data was processed as in Youngblut and colleagues[66], which consisted of read quality control and filtering of reads mapping to humans and other vertebrates. Quality controlled reads were assembled per-sample via metaSPAdes v.3.12.0 (ref. [68]). For mapping contigs to classified Methanobacteriales genomes, we selected species-level representative genomes from the Genome Taxonomy Database (GTDB; Release 95) based on optimal CheckM-estimated completeness and contamination ($n = 94$; Supplementary Table 12). Minimap2 v.2.20 was used to map all assembled contigs to all reference genomes, with an alignment length to query-length cut-off of ≥0.9. No contigs mapped to multiple references. All contigs mapping to any Methanothermobacteraceae or Methanothermobacteraceae A reference genome are available from FigShare (https://figshare.com/s/4c6ceb4ba8be4bab659f). Taxon relative abundances were estimated via Kraken2 v.2.1.1 and Bracken v.2.6.2 (ref. [69]), with a custom reference database constructed from the GTDB (Release 95) with Struo v.0.1.7 (refs. [70,71]). The database is available at http://ftp.tue.mpg.de/ebio/projects/struo/.

**Data analysis.** We used BLASTn[72] to assess similarity of ASVs to cultured representatives in the SILVA All Species Living Tree database v.132 (ref. [73]), with an E-value cut-off of $<1 \times 10^{-5}$. All BLAST hits with an alignment length <95% of the query sequence length were filtered out.

MRM was performed with the Ecodist v.2.0.5 R package[74]. We used rank-based correlations and 999 permutations to ascertain test significance. Regression variables that were not inherently distance matrices were converted via various means. Gower distance was used to convert detailed diet data, detailed habitat data and 'technicall' data (that is, captive/wild animal and faeces/gut contents sample type) to distance matrices. Geographic distance was calculated as great circle distance based on sample latitude and longitude. Alpha diversity was converted to a Euclidean distance matrix. PCoA ordinations were generated for each beta diversity measure via the Vegan v.2.5-6 R package[55].

Pagel's λ and LIPA were calculated via the Phylosignal v.1.3 R package[75], with 999 and 9,999 permutations, respectively. We tested for cophylogeny with the PACo) and ParaFit, implemented in the PACo v.0.4.2 (ref. [27]) and APE v.5.5 (ref. [26]) R packages, respectively. For both tests, the Cailliez correction[76] for negative eigenvalues was applied, and 999 permutations were used to assess significance. Tests of trait associations were performed with PGLS and RRPP, implemented in the phytools v.0.7-70 and RRPP v.0.6.2 packages[25], respectively. To ascertain significance, 999 permutations were used for both methods.

Ancestral state reconstruction models were fit to archaeal taxon relative abundances (extant traits) via the phylopars method as implemented in the Rphylopars package v.0.3.0[77]. The method incorporates intraspecies trait variation, so all samples were used instead of employing an intraspecies sensitivity analysis (above). We first compared log-likelihoods of four different models: Brownian

motion, Ornstein–Uhlenbeck, early burst and star phylogeny. Brownian motion and Ornstein–Uhlenbeck models had the best log-likelihoods for class- and genus-level archaeal relative abundances, respectively. Predicted trait values were visualized on the host phylogeny via the Phytools R package.

Supplementary Tables 7 and 8 list published body temperature and methane emission data used in this study.

Significant patterns of Archaea–Archaea and Archaea–Bacteria co-occurrence were inferred via the cooccur v.1.3 R package[78]. Subnetworks in each co-occurrence network were identified with the walktrap algorithm implemented in the igraph v.1.2.6 R package[79].

General data manipulation and visualization was performed in R[80] with the following R packages: dplyr v.1.0.1, tidyr v.1.1.0 and ggplot2 v.3.3.2 (ref. [81]). Phylogenies were manipulated and visualized with the APE and phytools R packages and with iTOL v.6.3 (ref. [82]). Networks were manipulated and visualized with the igraph v.1.2.6 (ref. [83]), tidygraph v.1.2.0 (ref. [84]) and ggraph v.2.0.4 (ref. [85]) R packages. High-performance computing cluster job submission was performed via the batchtools v.0.9.13 (ref. [86]) and clustermq v.0.8.95.1 (ref. [87]) R packages. For ASV-specific tests (*for example*, LIPA, PGLS, and co-occurrence), only ASVs present in >5% of samples were included. Multiple hypothesis testing was corrected via the Benjamini-Hochberg procedure.

**Statistics and reproducibility.** No statistical method was used to predetermine sample size. No data were excluded from the analyses, except for samples lacking sufficient 16S rRNA sequences (see the 16S RNA gene sequencing and data processing section above). The experiments were not randomized. The investigators were not blinded to allocation during experiments and outcome assessment. No statistical methods were used to predetermine sample sizes but our sample sizes are similar to those reported in previous publications[13,66]. Data met the assumptions of normality for instances in which parametric tests were used.

**Reporting Summary.** Further information on research design is available in the Nature Research Reporting Summary linked to this article.

## Data availability
The raw sequence data are available from the European Nucleotide Archive under the study accession numbers PRJEB40672 and PRJEB38078. All sample metadata used in this study is provided in Supplementary Table 1. The public databases used in this work include the Genome Taxonomy Database (https://gtdb.ecogenomic.org/), SILVA (https://www.arb-silva.de/), BLAST nr (https://ftp.ncbi.nlm.nih.gov/blast/) and Struo GTDB-r95 (http://ftp.tue.mpg.de/ebio/projects/struo/GTDB_release95/). Source data are provided with this paper.

## Code availability
All code and the software versions used for analyses are available at https://github.com/leylabmpi/16S-arc_vertebrate_paper. CoreGenomePrimers is available at https://github.com/leylabmpi/CoreGenomePrimers.

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

## Acknowledgements

This study was supported by the Max Planck Society and the Austrian Science Fund (FWF) research projects P23900 granted to A.H.F. and P22032 granted to G.H.R. Further support came from the Science Call 2015 'Resource und Lebensgrundlage Wasser' Project SC15-016 funded by the Niederösterreichische Forschungs- und Bildungsgesellschaft (NFB). We thank Andre Teare for providing body temperature data from the Medical Animal Records Keeping System (MedArks). We thank the following collaborators for their huge efforts in sample and data collection: Mario Baldi (School of Veterinary Medicine, Universidad Nacional de Costa Rica); Wolfgang Vogl and Frank Radon (Konrad Lorenz Institute of Ethology and Biological Station Illmitz); Endre Sós and Viktor Molnár (Budapest Zoo); Ulrike Streicher (Conservation and Wildlife Management Consultant, Vietnam); Katharina Mahr (Konrad Lorenz Institute of Ethology, University of Veterinary Medicine Vienna and Flinders University Adelaide, South Australia); Peggy Rismiller (Pelican Lagoon Research Centre, Australia); Rob Deaville (Institute of Zoology, Zoological Society of London); Alex Lécu (Muséum National d'Histoire Naturelle and Paris Zoo); Danny Govender and Emily Lane (South African National Parks, Sanparks); Fritz Reimoser (Research Institute of Wildlife Ecology, University of Veterinary Medicine Vienna); Anna Kübber-Heiss and Team (Pathology, Research Institute of Wildlife Ecology, University of Veterinary Medicine Vienna); Nikolaus Eisank (Nationalpark Hohe Tauern, Kärnten); Attila Hettyey and Yoshan Moodley (Konrad Lorenz Institute of Ethology, University of Veterinary Medicine Vienna); Mansour El-Matbouli and Oskar Schachner (Clinical Unit of Fish Medicine, University of Veterinary Medicine); Barbara Richter (Institute of Pathology and Forensic Veterinary Medicine, University of Veterinary Medicine Vienna); Hanna Vielgrader and Zoovet Team (Schönbrunn Zoo); Reinhard Pichler (Herberstein Zoo). We explicitly thank the Freek Venter of South African National Parks and the National Zoological Gardens of South Africa for granting access to their parks for sample collection.

## Author contributions

G.H.R., R.E.L. and A.H.F. created the study concept. G.H.R., C.W. and G.S. performed the sample collection and metadata compilation. G.H.R., S.M. and S.D. performed all molecular work. N.D.Y. performed the data analysis. N.D.Y. and R.E.L. wrote the manuscript. R.E.L. and N.D.Y. supervised the study.

## Funding

## Competing interests

The authors declare no competing interests.

## Additional information

**Correspondence and requests for materials** should be addressed to Nicholas D. Youngblut.

# Reporting Summary

## Statistics

For all statistical analyses, confirm that the following items are present in the figure legend, table legend, main text, or Methods section.

| n/a | Confirmed | |
|---|---|---|
| ☐ | ☒ | The exact sample size (*n*) for each experimental group/condition, given as a discrete number and unit of measurement |
| ☐ | ☒ | A statement on whether measurements were taken from distinct samples or whether the same sample was measured repeatedly |
| ☐ | ☒ | The statistical test(s) used AND whether they are one- or two-sided *Only common tests should be described solely by name; describe more complex techniques in the Methods section.* |
| ☐ | ☒ | A description of all covariates tested |
| ☐ | ☒ | A description of any assumptions or corrections, such as tests of normality and adjustment for multiple comparisons |
| ☐ | ☒ | A full description of the statistical parameters including central tendency (e.g. means) or other basic estimates (e.g. regression coefficient) AND variation (e.g. standard deviation) or associated estimates of uncertainty (e.g. confidence intervals) |
| ☐ | ☒ | For null hypothesis testing, the test statistic (e.g. *F*, *t*, *r*) with confidence intervals, effect sizes, degrees of freedom and *P* value noted *Give P values as exact values whenever suitable.* |
| ☒ | ☐ | For Bayesian analysis, information on the choice of priors and Markov chain Monte Carlo settings |
| ☒ | ☐ | For hierarchical and complex designs, identification of the appropriate level for tests and full reporting of outcomes |
| ☐ | ☒ | Estimates of effect sizes (e.g. Cohen's *d*, Pearson's *r*), indicating how they were calculated |

*Our web collection on statistics for biologists contains articles on many of the points above.*

## Software and code

Policy information about availability of computer code

| Data collection | All information on the data collection can be found in the manuscript (e.g., Methods and Supplementary Table 1) along with information at https://github.com/leylabmpi/16S-arc_vertebrate_paper. The following software was used: QIIME2 v2019.10, DADA2 v1.10.0, QIIME2 q2-feature-classifier v2019.10.0, iNEXT R package v2.0.19, zCompositions R package v1.3.3, MAFFT v7.310, metaSPAdes v3.12.0, minimap2 v2.20, Kraken2 v2.1.1, Bracken v2.6.2, Struo v0.1.7, Ecodist R package v2.0.5, Vegan R package v2.5-6 , phylosignal R package v1.3, PACo R package v0.4.2, APE R package v5.5, phytools R package v0.7-70, RRPP R package v0.6.2, Rphylopars R package v0.3.0,  cooccur R package v1.3, igraph R package v.1.2.6, iTOL 6.3, tidygraph R package v1.2.0, ggraph R package v2.0.4, batchtools R package v0.9.13, clustermq v0.8.95.1, dplyr R package v1.0.1,  tidyr R package v1.1.0, ggplot2 R package v3.3.2 |
|---|---|
| Data analysis | All code required to reproduce the analysis can be found at https://github.com/leylabmpi/16S-arc_vertebrate_paper |

For manuscripts utilizing custom algorithms or software that are central to the research but not yet described in published literature, software must be made available to editors and reviewers. We strongly encourage code deposition in a community repository (e.g. GitHub). See the Nature Portfolio guidelines for submitting code & software for further information.

## Data

Policy information about availability of data

All manuscripts must include a data availability statement. This statement should provide the following information, where applicable:
- Accession codes, unique identifiers, or web links for publicly available datasets
- A description of any restrictions on data availability
- For clinical datasets or third party data, please ensure that the statement adheres to our policy

Genome Taxonomy Database (https://gtdb.ecogenomic.org/)

PRJEB40672
PRJEB38078
SILVA (https://www.arb-silva.de/)
BLAST nr (https://ftp.ncbi.nlm.nih.gov/blast/)
Struo GTDB-r95 (http://ftp.tue.mpg.de/ebio/projects/struo/GTDB_release95/)

# Field-specific reporting

Please select the one below that is the best fit for your research. If you are not sure, read the appropriate sections before making your selection.

☐ Life sciences    ☐ Behavioural & social sciences    ☒ Ecological, evolutionary & environmental sciences

For a reference copy of the document with all sections, see nature.com/documents/nr-reporting-summary-flat.pdf

# Field work, collection and transport

| | |
|---|---|
| Field conditions | Samples were collected from many locations around the world at varying times of the year. See the supplemental metadata provided with the manuscript. Habitats include anthropogenic, cultivated, freshwater, grassland, saline water, terrestrial, and woodland. Ambient temperature and rainfall were not recorded. |
| Location | Sample=F14_Common_Bream,latitude=48.20833,longitude=16.373064,elevation=171m;Sample=F35_Red_Deer,latitude=48.20833,longitude=16.373064,elevation=171m;Sample=F36_Red_Deer,latitude=48.20833,longitude=16.373064,elevation=171m;Sample=F44_Fallow_Deer,latitude=48.20833,longitude=16.373064,elevation=171m;Sample=F45_Red_Deer,latitude=48.20833,longitude=16.373064,elevation=171m;Sample=F46_Red_Deer,latitude=48.20833,longitude=16.373064,elevation=171m;Sample=F47_Red_Deer,latitude=48.20833,longitude=16.373064,elevation=171m;Sample=F48_Red_Deer,latitude=48.20833,longitude=16.373064,elevation=171m;Sample=F53_Mouflon,latitude=48.20833,longitude=16.373064,elevation=171m;Sample=F66_Wild_Boar,latitude=48.20833,longitude=16.373064,elevation=171m;Sample=F68_Red_Deer,latitude=48.20833,longitude=16.373064,elevation=171m;Sample=F69_Red_Deer,latitude=48.20833,longitude=16.373064,elevation=171m;Sample=F70_Red_Deer,latitude=48.20833,longitude=16.373064,elevation=171m;Sample=F80_Red_Deer,latitude=48.20833,longitude=16.373064,elevation=171m;Sample=F90_Domestic_Dog,latitude=48.20833,longitude=16.373064,elevation=171m;Sample=X3_Alpine_Chamois,latitude=47.7779,longitude=13.23383,elevation=784m;Sample=X7_European_Otter,latitude=47.062,longitude=15.415,elevation=259m;Sample=X11_Onager,latitude=29.608056,longitude=52.524722,elevation=1544m;Sample=X13_Wolf,latitude=46.866667,longitude=14.116667,elevation=825m;Sample=X15_Tawny_Owl,latitude=48.20833,longitude=16.373064,elevation=171m;Sample=X16_Fat_Dormouse,latitude=48.20833,longitude=16.373064,elevation=171m;Sample=X22_European_Rabbit,latitude=48.20833,longitude=16.373064,elevation=171m;Sample=X23_Garden_Dormouse,latitude=48.20833,longitude=16.373064,elevation=171m;Sample=X31_Chicken,latitude=48.783333,longitude=15.066667,elevation=526m;Sample=X32_Greylag_Goose,latitude=48.783333,longitude=15.066667,elevation=526m;Sample=X33_Wild_Turkey,latitude=48.783333,longitude=15.066667,elevation=526m;Sample=X34_Mallard_Duck,latitude=48.783333,longitude=15.066667,elevation=526m;Sample=X40_Bactrian_Camel,latitude=43.707594,longitude=98.349609,elevation=1119m;Sample=X41_Pika,latitude=43.707594,longitude=98.349609,elevation=1119m;Sample=X42_Goitered_Gazelle,latitude=43.707594,longitude=98.349609,elevation=1119m;Sample=X43_European_Badger,latitude=48.20833,longitude=16.373064,elevation=171m;Sample=X60_European_Chub,latitude=48.2798373,longitude=15.411374,elevation=472m;Sample=F157a_European_Toad,latitude=48.333333,longitude=15.75,elevation=278m;Sample=X77_Onager,latitude=29.608056,longitude=52.524722,elevation=1544m;Sample=X85_Domestic_Dog,latitude=48.783333,longitude=15.066667,elevation=526m;Sample=X88_European_Rabbit,latitude=48.20833,longitude=16.373064,elevation=171m;Sample=X93_Red_Sheep,latitude=48.783333,longitude=15.066667,elevation=526m;Sample=X94_Mangalica,latitude=48.783333,longitude=15.066667,elevation=526m;Sample=X95_Meadow_Viper,latitude=47.5,longitude=19.05,elevation=104m;Sample=X101_Horse,latitude=48.783333,longitude=15.066667,elevation=526m;Sample=X102_Kulan,latitude=43.707594,longitude=98.349609,elevation=1119m;Sample=X109_Red_Fox,latitude=48.20833,longitude=16.373064,elevation=171m;Sample=X111_West_European_Hedgehog,latitude=47.5,longitude=19.05,elevation=104m;Sample=X116_Common_Kestrel,latitude=47.5,longitude=19.05,elevation=104m;Sample=X117_Long_eared_Owl,latitude=47.5,longitude=19.05,elevation=104m;Sample=X119_Ural_Owl,latitude=48.004722,longitude=15.167778,elevation=339m;Sample=X121_Raccoon_Dog,latitude=48.333333,longitude=15.75,elevation=278m;Sample=X122_Wild_Boar,latitude=47.84637,longitude=16.52796,elevation=168m;Sample=X123_Alpine_Chamois,latitude=47.04266,longitude=12.89106,elevation=1745m;Sample=X125_Common_Carp,latitude=48.7564,longitude=16.501147,elevation=200m;Sample=X127_Alpine_Marmot,latitude=47.0267064,longitude=12.7899939,elevation=1683m;Sample=X128_Alpine_Marmot,latitude=47.0267064,longitude=12.7899939,elevation=1683m;Sample=X129_Alpine_Ibex,latitude=47.066667,longitude=12.783333,elevation=2278m;Sample=X130_Alpine_Ibex,latitude=47.0855556,longitude=12.723333,elevation=2356m;Sample=X131_Rock_Ptarmigan,latitude=47.05,longitude=13.39999,elevation=2220m;Sample=X133_Domestic_Goat,latitude=47.5,longitude=16.416667,elevation=335m;Sample=X134_Domestic_Goat,latitude=47.5,longitude=16.416667,elevation=335m;Sample=X135_Horse,latitude=47.5,longitude=16.416667,elevation=335m;Sample=X157_Grey_Heron,latitude=48.333333,longitude=15.75,elevation=278m;Sample=X158_White_tailed_Eagle,latitude=48.33,longitude=16.868056,elevation=146m;Sample=X137_Gaur,latitude=11.423333,longitude=107.428611,elevation=120m;Sample=X138_Gaur,latitude=11.423333,longitude=107.428611,elevation=120m;Sample=X139_Sambar,latitude=11.423333,longitude=107.428611,elevation=120m;Sample=X140_Sambar,latitude=11.423333,longitude=107.428611,elevation=120m;Sample=X164_Greylag_Goose,latitude=54.516667,longitude=8.636944,elevation=minus2m;Sample=X170_Tawny_Owl,latitude=48.20833,longitude=16.373064,elevation=171m;Sample=X172_European_Greenfinch,latitude=47.5,longitude=19.05,elevation=104m;Sample=X174_Great_Tit,latitude=47.5,longitude=19.05,elevation=104m;Sample=X153_Bank_Vole,latitude=48.20833,longitude=16.373064,elevation=171m;Sample=X146_Cattle,latitude=50.59549,longitude=12.6386475,elevation=468m;Sample=X194_Agile_Frog,latitude=48.20833,longitude=16.373064,elevation=171m;Sample=X154_Tree_Shrew,latitude=48.20833,longitude=16.373064,elevation=171m;Sample=X141_Red_cheeked_Gibbon,latitude=11.423333,longitude=107.428611,elevation=120m;Sample=X179_Roach,latitude=48.236304,longitude=15.333137,elevation=207m;Sample=X195_Domestic_Cat,latitude=48.20833,longitude=16.373064,elevation=171m;Sample=X203_Red_Deer,latitude=47.0177778,longitude=12.780833,elevation=1868m;Sample=X206_Red_Sheep,latitude=47.726944,longitude=16.081667,elevation=367m;Sample=X212_Goose,latitude=47.5,longitude=16.416667,elevation=335m;Sample=X213_Goose,latitude=47.5,longitude=16.416667,elevation=335m;Sample=X214_Goose,latitude=47.5,longitude=16.416667,elevation=335m;Sample=X215_Goose,latitude=47.5,longitude=16.416667,elevation=335m;Sample=X218_Goose,latitude=47.5,longitude=16.416667,elevation=335m;Sample=X219_Goose,latitude=47.5,longitude=16.416667,elevation=335m;Sample=X247_Common_Hamster,latitude=48.410833,longitude=15.610278,elevation=195m;Sample=X252_Eurasian_Lynx,latitude=48.226667,longitude=14.179444,elevation=293m;Sample=X320_Red_Deer,latitude=48.20833,longitude=16.373064,elevation=171m;Sample=X233_Aesculapian_Snake,latitude=48.20833,longitude=16.373064,elevation=171m;Sample=X236_Pygmy_Slow_Loris,latitude=20.316667,longitude=105.608333,elevation=306m;Sample=X237_Red_shanked_Douc_Langur,latitude=20.316667,longitude=105.608333,elevation=306m;Sample=X238_Hanuman_Langur,latitude=20.316667,longitude=105.608333,elevation=306m;Sample=X240_Southern_White_cheeked_Gibbon,latitude=20.316667,longitude=105.608333,elevation=306m;Sample=X241_Italian_wall_lizard,latitude=45.25,longitude=15.466667,elevation=241m;Sample=X242_Dalmatian_Tortoise,latitude=45.25,longitude=15.466667,elevation=241m;Sample=X230_Carrion_Crow,latitude=48.333333,longitude=15.75,elevation=278m;Sample=X234_Beech_Marten,latitude=48.20833,longitude=16.373064,elevation=171m;Sample=X259_Mute_Swan,latitude=47.5,longitude=19.05,elevation=104m;Sample=X260_Blackbird,latitude=47.5,longitude=19.05,elevation=104m;Sample=X265_Western_Marsh_Harrier,latitude=48.333333,longitude=15.75,elevation=278m;Sample=X266_Western_Marsh_Harrier,latitude=48.333333,longitude=15.75,elevation=278m;Sample=X268_Rook,latitude=47.793889,longitude=16.495833,elevation=168m;Sample=X270_Common_Pheasant,latitude=48.477146,longitude=16.2079188,elevation=224m;Sample=X271_Common_Pheasant,latitude=48.477146,longitude=16.2079188,elevation=224m;Sample=X279_Western_Grey_Kangaroo,latitude=-35.833333,longitude=137.25,elevation=157m;Sample=X280_Western_Grey_Kangaroo,latitude=-35.833333,longitude=137.25,elevation=157m;Sample=X281_Tammar_Wallaby,latitude=-35.833333,longitude=137.25,elevation=157m;Sample=X283_Koala,latitude=-34.95,longitude=138.6833 |

# Ecological, evolutionary & environmental sciences study design

All studies must disclose on these points even when the disclosure is negative.

| | |
|---|---|
| Study description | We studied the gut microbiome of wild and captive animals. The major treatments were animal diet and evolutionary history. Other covariates included captivity status (wild vs captive) and sample type (gut contents or feces). Treatment factors and covariates include: host evolutionary history, host diet, host habitats, geographic distance between sampling locations, sample type (feces versus gut contents, and host captive/wild status. The dataset is hierarchically structured via host and microbial taxonomy and evolutionary history. The total number of experimental units (GI samples) is 311, but only 185 in the 16S rRNA sequence dataset. The replicates per treatment, when considering host species as treatments, varied from 1 to 12. |
| Research sample | The feces (or gut contents) from an individual animal was considered a single sample. We only used one sample per animal species for each hypothesis test, due to a lack of data on intra-species host evolutionary history. The gender ratio of individuals in which sex was known (n = 79) was 41:38 (m:f). The age range for the individual in which age is know is 2-10 years. |
| Sampling strategy | Samples were collected by wildlife biologists. No sample size calculation was performed. Sample sizes were chosen based on sample availability. The sample sizes were sufficient to decouple the association of host evolutionary history and diet with archaeal diversity, which is the major goal of this work. |
| Data collection | Sampling metadata was collected by sampling teams at the respective locations and reported by email to the study team who aggregated metadata in spreadsheets, adding appropriate additional data from literature sources and databases (diet data, host taxonomy data, etc.). Sample data was collected by the following persons: Mario Baldi, School of Veterinary Medicine, Universidad Nacional de Costa Rica; Wolfgang Vogl and Frank Radon, Konrad Lorenz Institute of Ethology and Biological Station Illmitz; Endre Sós and Viktor Molnár, Budapest Zoo; Ulrike Streicher, Conservation and Wildlife Management Consultant, Vietnam; Katharina Mahr, Konrad Lorenz Institute of Ethology, University of Veterinary Medicine Vienna and Flinders University Adelaide, South Australia; Peggy Rismiller, Pelican Lagoon Research Centre, Australia; Rob Deaville, Institute of Zoology, Zoological Society of London; Alex Lécu, Muséum National d'Histoire Naturelle and Paris Zoo; Danny Govender and Emily Lane, South African National Parks, Sanparks; Fritz Reimoser, Research Institute of Wildlife Ecology, University of Veterinary Medicine Vienna; Anna Kübber-Heiss and Team, Pathology, Research Institute of Wildlife Ecology, University of Veterinary Medicine Vienna; Nikolaus Eisank, Nationalpark Hohe Tauern, Kärnten; Attila Hettyey and Yoshan Moodley, Konrad Lorenz Institute of Ethology, University of Veterinary Medicine Vienna; Mansour El-Matbouli and Oskar Schachner, Clinical Unit of Fish Medicine, University of Veterinary Medicine; Barbara Richter, Institute of Pathology and Forensic Veterinary Medicine, University of Veterinary Medicine Vienna; Hanna Vielgrader and Zoovet Team, Schönbrunn Zoo; Reinhard Pichler, Herberstein Zoo. Freek Venter, South African National Parks and the National Zoological Gardens of South Africa. DNA concentrations were measured using a PicoGreen reagent (Thermo-Fisher, Vienna, Austria) using a Anthos Zenyth fluorescence plate reader (Biochrom, Cambridge UK). |
| Timing and spatial scale | Sampling was conducted from February 2009 and March 2014. Samples originated predominantly from Central Europe (Austria and neighboring countries). However, in order to cover as much vertebrate diversity as possible, many samples were also taken from other countries around the world (see the metadata provided with the manuscript). The frequency and periodicity of sampling was based on sample manpower availability. |
| Data exclusions | A small subset of samples were excluded from all analyses of the 16S rRNA sequence data due to not enough sampling depth of the microbiome. These exclusion criteria were not pre-established, but such exclusions are standard for microbiome data. |
| Reproducibility | We assessed each question with multiple analyses and compared our results to previous studies in order to assess reproducibility. Experimental replication was not performed. |
| Randomization | We utilize a sensitivity method of randomly subsampling one individual sample per species for each hypothesis test, repeating this procedure a total of 100 times, and using the 95% quartile of significance values for each individual subsample to assess overall significance. This allowed us to assess how sensitive our analysis was to intra-species heterogeneity, which would be missed if we had simply used one randomly subsample sample per animal species. |
| Blinding | No blinding was used, given that this study is not set up like a clinical trial; we are not testing control versus treatment as done with a drug trial, or similar. |

Did the study involve field work?    ☒ Yes    ☐ No

33,elevation=388m;Sample=X285_Short_beaked_Echidna,latitude=-35.833333,longitude=137.25,elevation=157m;Sample=X286_Short_beaked_Echidna,latitude=-35.833333,longitude=137.25,elevation=157m;Sample=X287_Common_Brushtail,latitude=-34.929,longitude=138.601,elevation=45m;Sample=X288_Common_Brushtail,latitude=-34.929,longitude=138.601,elevation=45m;Sample=X289_Eastern_Grey_Kangaroo,latitude=-30.000232,longitude=136.209155,elevation=132m;Sample=X290_Eastern_Grey_Kangaroo,latitude=-30.000232,longitude=136.209155,elevation=132m;Sample=X292_Central_Bearded_Dragon,latitude=-35.63,longitude=138.503611,elevation=69m;Sample=X305_European_Hare,latitude=48.7564,longitude=16.501147,elevation=200m;Sample=X326_Silver_Bream,latitude=48.721773,longitude=16.193848,elevation=190m;Sample=X327_Silver_Bream,latitude=48.721773,longitude=16.193848,elevation=190m;Sample=X328_Prussian_Carp,latitude=48.721773,longitude=16.193848,elevation=190m;Sample=X330_Ide,latitude=48.721773,longitude=16.193848,elevation=190m;Sample=X333_Gray_Seal,latitude=54.958004,longitude=-1.351318,elevation=0m;Sample=X336_Sei_Whale,latitude=55.289869,longitude=-1.564178,elevation=0m;Sample=X338_White_beaked_Dolphin,latitude=51.395136,longitude=1.380501,elevation=0m;Sample=X339_White_beaked_Dolphin,latitude=52.375861,longitude=1.712236,elevation=0m;Sample=X340_Western_Lowland_Gorilla,latitude=0.219726,longitude=14.853516,elevation=414m;Sample=X341_Western_Lowland_Gorilla,latitude=0.219726,longitude=14.853516,elevation=414m;Sample=X344_Indian_Rhinoceros,latitude=27.529131,longitude=84.354205,elevation=197m;Sample=X349_Smooth_Newt,latitude=48.333333,longitude=15.75,elevation=278m;Sample=X350_Smooth_Newt,latitude=48.333333,longitude=15.75,elevation=278m;Sample=X351_Grass_Snake,latitude=48.20833,longitude=16.373064,elevation=171m;Sample=X352_Kinkajou,latitude=12.822514,longitude=-84.100342,elevation=45m;Sample=X353_Raccoon,latitude

=9.96304,longitude=-84.04823,elevation=1237m;Sample=X356_Northern_Tamandua,latitude=6.489983,longitude=-75.19043,elevation=1980m;Sample=X357_Northern_Tamandua,latitude=6.489983,longitude=-75.19043,elevation=1980m;Sample=X359_Brown_throated_Sloth,latitude=12.822514,longitude=-84.100342,elevation=45m;Sample=X360_White_tailed_Deer,latitude=9.748917,longitude=-83.753428,elevation=1403m;Sample=X363_Desmarests_Spiny_Pocket_Mouse,latitude=9.748917,longitude=-83.753428,elevation=1403m;Sample=X368_European_Ground_Squirrel,latitude=48.20833,longitude=16.373064,elevation=171m;Sample=X369_Koala,latitude=48.20833,longitude=16.373064,elevation=171m;Sample=X370_African_Bush_Elephant,latitude=48.20833,longitude=16.373064,elevation=171m;Sample=X371_Linnaeus_Two_toed_Sloth,latitude=48.20833,longitude=16.373064,elevation=171m;Sample=X372_Emu,latitude=48.20833,longitude=16.373064,elevation=171m;Sample=X376_Kulan,latitude=45.4,longitude=92.9,elevation=1345m;Sample=X378_Fat_Sand_Rat,latitude=48.20833,longitude=16.373064,elevation=171m;Sample=X380_Wild_Boar,latitude=47.5,longitude=16.416667,elevation=335m;Sample=X381_Wild_Boar,latitude=47.5,longitude=16.416667,elevation=335m;Sample=X382_Arctic_Wolf,latitude=47.224,longitude=15.812,elevation=423m;Sample=X384_Eurasian_Lynx,latitude=47.224,longitude=15.812,elevation=423m;Sample=X389_Indian_Rhinoceros,latitude=48.20833,longitude=16.373064,elevation=171m;Sample=X390_European_Ground_Squirrel,latitude=48.20833,longitude=16.373064,elevation=171m;Sample=X391_European_Ground_Squirrel,latitude=48.2276,longitude=14.024048,elevation=354m;Sample=X394_Przewalski_horse,latitude=45.4,longitude=92.9,elevation=1345m;Sample=X395_Przewalski_horse,latitude=47.224,longitude=15.812,elevation=423m;Sample=X396_Morelets_crocodile,latitude=48.20833,longitude=16.373064,elevation=171m;Sample=X403_Chacma_Baboon,latitude=-25.119855,longitude=31.91541,elevation=176m;Sample=X404_Brown_Greater_Galago,latitude=-24.995833,longitude=31.591944,elevation=288m;Sample=X405_Nile_Crocodile,latitude=-23.231251,longitude=30.492554,elevation=510m;Sample=X407_African_Bush_Elephant,latitude=-24.995833,longitude=31.591944,elevation=288m;Sample=X408_African_Bush_Elephant,latitude=-24.995833,longitude=31.591944,elevation=288m;Sample=X409_Impala,latitude=-24.995833,longitude=31.591944,elevation=288m;Sample=X410_Impala,latitude=-24.995833,longitude=31.591944,elevation=288m;Sample=X411_Leopard,latitude=-24.439148,longitude=31.438065,elevation=418m;Sample=X414_Vervet_Monkey,latitude=-24.995833,longitude=31.591944,elevation=288m;Sample=X415_White_Rhinoceros,latitude=-23.231251,longitude=30.492554,elevation=510m;Sample=X416_White_Rhinoceros,latitude=-23.231251,longitude=30.492554,elevation=510m;Sample=X417_Giraffe,latitude=1.230374,longitude=38.583984,elevation=345m;Sample=X418_Giraffe,latitude=1.230374,longitude=38.583984,elevation=345m;Sample=X423_Quagga,latitude=-2.416276,longitude=34.650879,elevation=1527m;Sample=X424_Quagga,latitude=-2.416276,longitude=34.650879,elevation=1527m;Sample=X421_Common_Hippopotamus,latitude=-2.416276,longitude=34.650879,elevation=1527m;Sample=X422_Common_Hippopotamus,latitude=-2.416276,longitude=34.650879,elevation=1527m;Sample=X419_African_Buffalo,latitude=-2.416276,longitude=34.650879,elevation=1527m;Sample=X420_Cattle,latitude=-2.416276,longitude=34.650879,elevation=1527m;Sample=X426_Ural_Owl,latitude=48.20833,longitude=16.373064,elevation=171m;Sample=X427_Great_Spotted_Woodpecker,latitude=48.20833,longitude=16.373064,elevation=171m;Sample=X428_African_Bush_Elephant,latitude=48.20833,longitude=16.373064,elevation=171m;Sample=X429_Giraffe,latitude=48.20833,longitude=16.373064,elevation=171m;Sample=X430_Giraffe,latitude=48.20833,longitude=16.373064,elevation=171m;Sample=X431_Quagga,latitude=47.224,longitude=15.812,elevation=423m;Sample=X432_Vervet_Monkey,latitude=47.224,longitude=15.812,elevation=423m;Sample=X434_Blue_Wildebeest,latitude=47.224,longitude=15.812,elevation=423m;Sample=X435_Bactrian_Camel,latitude=47.224,longitude=15.812,elevation=423m;Sample=X66_Mouflon,latitude=48.20833,longitude=16.373064,elevation=171m;Sample=X221_Alpine_Ibex,latitude=47.7594088,longitude=13.060725,elevation=436m;Sample=X73_One_humped_Camel,latitude=48.214922,longitude=16.936046,elevation=158m;Sample=X67_European_Roe,latitude=48.20833,longitude=16.373064,elevation=171m;Sample=X96_European_Hare,latitude=47.897778,longitude=16.908889,elevation=126m

**Access & import/export**

Wildlife biologists, who were conducting long-term research on the respective species in its habitat, ensured that sampling guidelines and restrictions were adhered to, where these were applicable. Samples were collected by the following persons:
Mario Baldi, School of Veterinary Medicine, Universidad Nacional de Costa Rica; Wolfgang Vogl and Frank Radon, Konrad Lorenz Institute of Ethology and Biological Station Illmitz; Endre Sós and Viktor Molnár, Budapest Zoo; Ulrike Streicher, Conservation and Wildlife Management Consultant, Vietnam; Katharina Mahr, Konrad Lorenz Institute of Ethology, University of Veterinary Medicine Vienna and Flinders University Adelaide, South Australia; Peggy Rismiller, Pelican Lagoon Research Centre, Australia; Rob Deaville, Institute of Zoology, Zoological Society of London; Alex Lécu, Muséum National d'Histoire Naturelle and Paris Zoo; Danny Govender and Emily Lane, South African National Parks, Sanparks; Fritz Reimoser, Research Institute of Wildlife Ecology, University of Veterinary Medicine Vienna; Anna Kübber-Heiss and Team, Pathology, Research Institute of Wildlife Ecology, University of Veterinary Medicine Vienna; Nikolaus Eisank, Nationalpark Hohe Tauern, Kärnten; Attila Hettyey and Yoshan Moodley, Konrad Lorenz Institute of Ethology, University of Veterinary Medicine Vienna; Mansour El-Matbouli and Oskar Schachner, Clinical Unit of Fish Medicine, University of Veterinary Medicine; Barbara Richter, Institute of Pathology and Forensic Veterinary Medicine, University of Veterinary Medicine Vienna; Hanna Vielgrader and Zoovet Team, Schönbrunn Zoo; Reinhard Pichler, Herberstein Zoo. Freek Venter, South African National Parks and the National Zoological Gardens of South Africa. Most of the sampling did not require permits, since only non-invasive, fecal samples were collected. The City of Vienna issued a permit for the capture and sampling wild mice (issuing authority: Municipal department 22 of the City of Vienna, date of issue April 6 2011, permit number MA 22 - 229/2011). The South African National Parks organization issued a permit for collecting and exporting fecal material from the Park grounds (issuing authority: South African National Parks, date of issue November 18th 6 2013, reference number REISHG 1158)

**Disturbance**

No disturbance was caused.

# Reporting for specific materials, systems and methods

We require information from authors about some types of materials, experimental systems and methods used in many studies. Here, indicate whether each material, system or method listed is relevant to your study. If you are not sure if a list item applies to your research, read the appropriate section before selecting a response.

## Materials & experimental systems

| n/a | Involved in the study |
|-----|----------------------|
| ☒ | Antibodies |
| ☒ | Eukaryotic cell lines |
| ☒ | Palaeontology and archaeology |
| ☒ | Animals and other organisms |
| ☒ | Human research participants |
| ☒ | Clinical data |
| ☒ | Dual use research of concern |

## Methods

| n/a | Involved in the study |
|-----|----------------------|
| ☒ | ChIP-seq |
| ☒ | Flow cytometry |
| ☒ | MRI-based neuroimaging |

