## [Peer Review File · Nature Microbiology]

Peer Review Information

Journal: Nature Microbiology

Manuscript Title:

Vertebrate host phylogeny influences gut archaeal diversity

Corresponding author name(s): Nicholas Youngblut

Reviewer Comments & Decisions:

Decision Letter, initial version:

Dear Nick,

Thank you for your patience while your manuscript "Strong influence of vertebrate host phylogeny on gut archaeal diversity" was under peer-review at Nature Microbiology. It has now been seen by 3 referees, whose expertise and comments you will find at the of this email. Although they find your work of some potential interest, they have raised a number of concerns that will need to be addressed before we can consider publication of the work in Nature Microbiology.

In particular, all referees have concerns with the sample size and number of replicates, and therefore how robust your conclusions are. The referees suggest using qPCR or other genomic approaches to validate your findings and provide more insight into particular groups e.g. Methanothermobacter, as well as providing more information on the failed samples i.e. low quality DNA or low abundance of archaea. Other concerns include the use of negative controls and further analyses on ASVs shared between samples.

Should further experimental data allow you to address these criticisms, we would be happy to look at a revised manuscript.

We strongly support public availability of data. Please place the data used in your paper into a public data repository, if one exists, or alternatively, present the data as Source Data or Supplementary Information. If data can only be shared on request, please explain why in your Data Availability Statement, and also in the correspondence with your editor. For some data types, deposition in a

public repository is mandatory - more information on our data deposition policies and available repositories can be found at <https://www.nature.com/nature-research/editorial-policies/reporting-standards#availability-of-data>.

Please include a data availability statement as a separate section after Methods but before references, under the heading "Data Availability". This section should inform readers about the availability of the data used to support the conclusions of your study. This information includes accession codes to public repositories (data banks for protein, DNA or RNA sequences, microarray, proteomics data etc...), references to source data published alongside the paper, unique identifiers such as URLs to data repository entries, or data set DOIs, and any other statement about data availability. At a minimum, you should include the following statement: "The data that support the findings of this study are available from the corresponding author upon request", mentioning any restrictions on availability. If DOIs are provided, we also strongly encourage including these in the Reference list (authors, title, publisher (repository name), identifier, year). For more guidance on how to write this section please see:

<http://www.nature.com/authors/policies/data/data-availability-statements-data-citations.pdf>

* If you have not done so already we suggest that you begin to revise your manuscript so that it conforms to our Article format instructions at <http://www.nature.com/nmicrobiol/info/final-submission>. Refer also to any guidelines provided in this letter.

When submitting the revised version of your manuscript, please pay close attention to our [href="https://www.nature.com/nature-research/editorial-policies/image-integrity">Digital Image Integrity Guidelines.](https://www.nature.com/nature-research/editorial-policies/image-integrity) and to the following points below:

-- that unprocessed scans are clearly labelled and match the gels and western blots presented in figures.

-- that control panels for gels and western blots are appropriately described as loading on sample processing controls

-- all images in the paper are checked for duplication of panels and for splicing of gel lanes.

{REDACTED}

Note: This url links to your confidential homepage and associated information about manuscripts you may have submitted or be reviewing for us. If you wish to forward this e-mail to co-authors, please delete this link to your homepage first.

Nature Microbiology is committed to improving transparency in authorship. As part of our efforts in this direction, we are now requesting that all authors identified as 'corresponding author' on published papers create and link their Open Researcher and Contributor Identifier (ORCID) with their account on the Manuscript Tracking System (MTS), prior to acceptance. This applies to primary research papers only. ORCID helps the scientific community achieve unambiguous attribution of all scholarly contributions. You can create and link your ORCID from the home page of the MTS by clicking on 'Modify my Springer Nature account'. For more information please visit www.springernature.com/orcid.

If you wish to submit a suitably revised manuscript we would hope to receive it within 6 months. If you cannot send it within this time, please let us know. We will be happy to consider your revision, even if a similar study has been accepted for publication at Nature Microbiology or published elsewhere (up to a maximum of 6 months).

{REDACTED}

Reviewer Expertise:

Referee #1: archaea, sequencing

Referee #2: vertebrate microbiome, evolution

Referee #3: human microbiome, microbial ecology, sequencing

Reviewer Comments:

Reviewer #1 (Remarks to the Author):

In their manuscript, Youngblut et al describe the correlation of vertebrate host phylogeny and archaeal diversity observed in gastrointestinal samples.

The entire study is based on a previous publication from 2019, describing the microbiome of various vertebrates and its correlation with phylogeny and diet. In this new manuscript, the authors concentrate on the archaeal community by analysing archaea-specific amplicons of the previously taken samples.

The dataset is highly interesting and provides a number of novel insights given that not much is known on the archaeal diversity of vertebrates in general.

However, the information gathered is based on amplicon information only, and does not explore

identified correlations further, by e.g. analysing quantitative or genomic information (e.g. for *Methanothermobacter*).

Also, the study is suffering from two major short-comings: Firstly, only 60% of all samples were positive for archaeal amplification. This leads to problems to consider certain vertebrate groups at all (e.g. humans could obviously not be included, as all samples from humans failed), or reduces the number of available replicates substantially (e.g. one blackbird sample failed, the other passed). This probably affects the conclusions drawn, as also, the conclusions were not further confirmed by any other means, which I strongly suggest. E.g. you could use *Methanothermobacter*-specific qPCR to confirm its association to aves. Furthermore, it would be interesting to explore more on the *Methanobrevibacter* diversity, as this species is obviously a very important host-associated archaeon. Secondly, negative controls are not mentioned. It remains unclear, whether process controls were performed and if so, how the information from negative controls was processed. This is important, as potential contamination would strongly affect the results and conclusions as the archaeal diversity observed is very low (differently to bacterial diversity).

At the moment of reviewing this manuscript, the underlying data were not accessible (BioProject, Github resource).

Additional specific comments:

The discussion is somewhat lengthy compared to the introduction, which was kept very short.

Line 22: gastrointestinal samples ? (maybe explain in more detail afterwards what is the difference of fecal and gut samples)

Line 26: *Methanothermobacter* should be written in italics, please check also elsewhere: all genus and species names should be written in italics

Line 26: "enriched" is not correct here, as enrichment would mean actual numbers, its only relative information

Line 30: "common ancestor" (not command)

Line 35: rephrase "waste products"

Line 37: 16S rRNA GENE sequencing (please correct throughout); likewise, please correct: 16S rRNA gene primers

Line 74: please explain, what do you mean with "adequate sequence data"

Fig. 1: How was the cladogram (diet) prepared? Colors should be adapted (captive/wild in 1A and 1D; Diet: 1A and 1C, etc...)

Line 109: more precise information?

Fig. 2: 2A consider to reduce information to one specific measure; Explain bracket without stars in Fig. 2E

Line 202: explain CI

Line 276: please explain in more detail what is shown in the sub-networks.

In general, please expand a bit on the sampling of wild animals to give the reader more background information without having to read the previous work.

Also, please be careful when using "abundances" to state that you refer to relative abundances. (e.g. lines 216 ff)

Fig. 3: the color code used in the trees should be explained in the figure itself.

Line 245: maybe the finding that *Methanothermobacter* is associated with higher temperature is only due to its association to Aves?

Line 248: which evidence do you mean?

Line 252: This requires a bit more support, such as qPCR, reconstructed genome fragments...

Line 265: in my opinion, the archaea-bacteria interaction/co-correlation could be analysed more deeply. Which archaea-bacteria co-occurrences did you observe (species, genera...)

Discussion: Consider to include a limitations chapter.

Line 408: Materials and Methods: Silva 119 is already quite old and a more up-to-date version should be used

Reviewer #2 (Remarks to the Author):

This paper is a nice examination of archaeal diversity across 110 vertebrate species. It is well written and the analyses are appropriate and effectively utilized. Although the paper is clearly very descriptive, given how little we know about archaea in vertebrate hosts, the data represent a nice contribution to the literature.

That said, I do have an important concern about the study with regard to sample size. The authors state that, after samples were removed due to failure to amplify and pass quality filtering standards, they included 185 samples from 110 host species into the analysis. While this is a large number of host species, these numbers indicate that there are very few species for which multiple individuals were sampled/analyzed. In fact, it seems the majority of host species are likely to be represented by a single individual. In my mind, this sampling distribution has the potential to invalidate many of the analyses, particularly those that relate to host phylogeny. For example, for ancestral state reconstruction, can we really be sure if these microbes are present/absent in a host species if we've only sampled one individual? While none of the analyses go as far as the host species level, I am still concerned that analyses examining higher levels of host taxonomy could be biased by this low coverage sampling. An important question would be how robust the results are to reducing the dataset to one in which at least five individuals per host species are sampled? Even this seems like low coverage, but is much better than one or two individuals. That said, I am not sure if most of the analyses performed are even possible on a dataset like that since I think it would reduce the number of host taxa represented substantially...

I was also interested in knowing more about the samples that did not amplify. Although I understand the authors' argument that samples that do not amplify or pass quality filtering could have either very low abundances of archaea or low quality DNA, it seems to me that they might be able to distinguish between these possibilities rather than vaguely speculating. Since the same samples were used for bacterial analyses in a previous paper, it would be possible to compare the number of samples that failed in both analyses and match samples to determine if the same samples failed both assays or not. For samples that failed both, it seems like that the DNA quantity/quality was low. For samples that passed the bacterial analysis but failed the archaeal analysis, it is more likely that the archaea are low abundance/less prevalent across the sample set. Doing this comparison and briefly discussing it (and/or altering any conclusions accordingly) would strengthen the paper in my opinion.

A few minor detailed comments from the text:

Line 169-171: It seems like this section or results should go above with the other alpha and beta diversity results despite relying on the same analysis described here.

Line 335: I am understanding that the authors are saying methanogens are less likely to vary with host diet here, but weren't methanogens flagged as the subset of ASVs being driven by diet earlier in the manuscript?

Methods: Information about sequencing depth overall and per sample would be helpful to include.

Reviewer #3 (Remarks to the Author):

Overall I love this paper, strong statistical analysis and compelling findings.

Ln 73-75 – it is possible this is due to variable sample size obtained from different animals? While it appears as though biological replication was performed, it was variable across the species tested, is this because replicates failed? Was any attempt made to associate failure with DNA concentration obtained, as normalized by biomass extracted?

It is not exactly clear to me whether a single ASVs was observed to be shared between species, or whether such sharing was not observed? e.g. Batharchaeia ASVs were also distributed differently between different species, right?

“Of 140 samples that overlap between our Archaea-targeted 16S dataset (“16S-arc”) and that from our previous work with standard “universal” 16S primers (“16S-uni”), 1390 versus only 169 archaeal ASVs were observed in each respective dataset (Figure S7)” – is this just referring to the number of ASVs or the number of ASVs that overlap (is that even possible? i.e. did you see ASVs that were identical across the datasets with different primers? I couldn’t find that analysis in the supplementary methods, but I may have missed it).

Would have been nice to get the qPCR for these primers across all samples? Is that still possible? This influences whether you use ASV abundances (as used currently), versus ASV proportions, which is more appropriate here.

Fig S8 ordination plots – what explains the strange ordination for weighted matrices? It looks like there are two different gradients, and maybe each is driven by proportions of one sequence/taxon, driving the data in that bifurcated arrow like pattern?

Ln128-129 – did you run it for just mammalian species as well?

Author Rebuttal to Initial comments

Reviewer #1 (Remarks to the Author)

Major comments

In their manuscript, Youngblut et al describe the correlation of vertebrate host phylogeny and archaeal diversity observed in gastrointestinal samples.

The entire study is based on a previous publication from 2019, describing the microbiome of various vertebrates and its correlation with phylogeny and diet. In this new manuscript, the authors concentrate on the archaeal community by analysing archaea-specific amplicons of the previously taken samples.

The dataset is highly interesting and provides a number of novel insights given that not much is known on the archaeal diversity of vertebrates in general.

However, the information gathered is based on amplicon information only, and does not explore identified correlations further, by e.g. analysing quantitative or genomic information (e.g. for *Methanothermobacter*).

Also, the study is suffering from two major short-comings: Firstly, only 60% of all samples were positive for archaeal amplification. This leads to problems to consider certain vertebrate groups at all (e.g. humans could obviously not be included, as all samples from humans failed), or reduces the number of available replicates substantially (e.g. one blackbird sample failed, the other passed). This probably affects the conclusions drawn, as also, the conclusions were not further confirmed by any other means, which I strongly suggest. E.g. you could use *Methanothermobacter*-specific qPCR to confirm its association to aves. Furthermore, it would be interesting to explore more on the *Methanobrevibacter* diversity, as this species is obviously a very important host-associated archaeon.

Response: The broad taxonomic scope of our analyses (*i.e.*, 5 taxonomic classes of vertebrates) allowed us to leverage inter-species evolutionary and ecological relatedness in order to produce robust conclusions about broad-scale patterns of archaeal community diversity in the vertebrate gut. Our general approach is akin to regression-based experimental design with few per-treatment data points, rather than an ANOVA-style approach with fewer treatments but more replicates per treatment (see also our response to Reviewer 2). To illustrate, although we may have just one sample for a particular host species and so no assessment of intra-species geographic variation, we can leverage inter-species variation in geographic locations across broad taxonomic groups in order to assess the influence of geography. We specifically did not include analyses targeted at finely-resolved host clades (*e.g.*, archaeal diversity in primates), given that we could not leverage as much inter-species variance due to the reduced number of host species. While a larger dataset comprising more host species and samples per species would almost definitely provide even more robust and resolved findings, we note that our study is the largest reported *Archaea*-targeted survey of vertebrate gut microbiome diversity, especially when considering that 76% of the individuals in the study are wild. We have updated the Introduction and Discussion to emphasize these points.

As suggested by the reviewer, we have added qPCR data supporting our initial findings that *Methanothermobacter* is present in many avian and mammalian species. Due to the lack of published highly specific qPCR primers for *Methanothermobacter*, we developed our own software pipeline for generating clade-specific qPCR primers targeting single copy core genes of any pan-genome (<https://github.com/levlabmpi/CoreGenomePrimers>). The workflow of the method is shown in Figure S26A and described in detail in the Supplemental Methods. The software is straight-forward to use with good documentation, scales to large datasets (*e.g.*, we have now used it to design qPCR primers for many microbial clades), and the code is freely available with a liberal open source license.

We used the primer design software to generate novel *Methanothermobacter*-specific primers based on publicly available genomes ($n = 16$). The primers ("174FR") target a gene cluster annotated as a 30S ribosomal protein, and BLAST searches of this gene versus NCBI nr showed that the closest non-target sequence shared only ~80% sequence identity to the target gene cluster. The primers include degeneracies to target all *Methanothermobacter* genomes, but still the primers are fully conserved (no degeneracies) at the 3' ends (the last 5 base pairs). Screening the primers against a panel of gDNA from various methanogen and bacterial isolates showed that the primers were indeed specific to *Methanothermobacter* (Figure S26B). Many mammalian and avian samples lacked any usable gDNA for qPCR, but we were able to conduct a 3x technical replicate qPCR assay with many avian and mammalian samples (Figure S27). The results supported our initial findings that *Methanothermobacter* is present in many avian and mammalian species. Notably, *Methanothermobacter* detection was consistent among members of the same species (*i.e.*, always present or always absent). The qPCR data did not show as widespread of *Methanothermobacter* prevalence as was seen for the 16S rRNA sequence data (Figure S24), but this may be due to the low amount of animal GI sample gDNA available for the assay, or possibly due to primer biases limiting the amount of *Methanothermobacter* diversity detected by the qPCR primers (*i.e.*, the primers we designed from existing genomes, which may be rather divergent from host-associated *Methanothermobacter*). Due to the smaller sample size of the qPCR assay relative to the 16S rRNA sequence data, and the lack of overlapping host body temperature data with the qPCR findings, we could not use these additional data to support or refute the hypothesis that host body temperature modulates *Methanothermobacter* abundance (Figure S27). Still, we feel that the 16S rRNA sequence data provides compelling evidence that host body temperature modulates *Methanothermobacter* and possibly *Methanobrevibacter* abundance. We have updated the text in the Results, Discussion, Supplemental Methods, and Supplemental Results to include these new findings.

Also as Reviewer 1 suggested, we utilized assembled contigs from metagenomes of the same animal GI sample collection to investigate the distribution of *Methanothermobacter* among host species. Specifically, we utilized the existing published metagenome sequence data from the same vertebrate GI sample dataset (DOI: 10.1128/mSystems.01045-20) and also more deeply sequenced 60 avian samples in hopes of detecting *Methanothermobacter* and possibly assembling an acceptably complete genome (*i.e.*, $\geq 50\%$ CheckM-estimated completeness). The metagenome assembly procedure was as reported previously and is included in the Supplemental Methods (DOI: 10.1128/mSystems.01045-20). We were able to assemble thousands of contigs (≥ 1.5 kb) that mapped solely to *Methanothermobacter* reference genomes (Figure S28). Across the 60 avian samples (41 species), 152 ± 1336 contigs mapped to *Methanothermobacter* reference genomes, which is 0.88 ± 2.2 (s.d.) percent of the total assembled contigs. Thus, the findings support our hypothesis that *Methanothermobacter* is prevalent across avian samples. We note that further extending our metagenome-based analysis to full assessments of archaeal community diversity is hampered by the low abundance of *Archaea* relative to *Bacteria* (and often large proportions of host DNA in the metagenomes) and the limited number of closely related reference genomes (see DOI: 10.1128/mSystems.01045-20). Moreover, such an extensive metagenomic analysis would extend this work beyond the scope of one study.

In regards to including more investigations of *Methanobrevibacter* diversity, we have added another figure that shows how *Methanobrevibacter* comprised a few dominant, closely related strains with varying distributions across host taxonomy and diet (Figure S9). This analysis, coupled with our findings that i) particular *Methanobrevibacter* ASVs significantly associate with host diet (Figure S16) or evolutionary history (Figure S17)

and ii) particular *Methanobrevibacter* ASVs co-occur with specific *Archaea* and *Bacteria* (Figures 4 & S30) both highlight potential niche partitioning among closely related *Methanobrevibacter* taxa. We have updated the Results and Discussion to include our new results (*i.e.*, Figure S9) and highlight this possible niche partitioning among closely related *Methanobrevibacter* taxa.

Secondly, negative controls are not mentioned. It remains unclear, whether process controls were performed and if so, how the information from negative controls was processed. This is important, as potential contamination would strongly affect the results and conclusions as the archaeal diversity observed is very low (differently to bacterial diversity).

Response: We thank the reviewer for pointing out this omission; we failed to convey this important initial step in the methodological details of the manuscript. We did include a number of negative controls ($n = 10$). We always compared PCR product production for all samples via measuring fluorescence during the PCR. This was done by adding a small amount of SYBR Green to each reaction and then measuring fluorescence via a Bio-Rad CFX qPCR thermocycler. We did not observe appreciable fluorescence in the negative controls relative to the treatment samples. Still, we generated Illumina libraries for all negative controls. While each negative control did generate ASVs (Illumina sequencing of negative controls always generates some viable sequence data, in our experience), the total ASV abundance and number of ASVs was orders of magnitude lower than the treatment samples (see Figure S5). These results indicate that if there was any contamination, the contribution to treatment sample community compositions would be negligible, especially given that rarefying removed any treatment samples with low numbers of sequences (see the Supplemental Methods). Importantly, the negative controls were dominated by 6 Gram positive bacterial genera: *Catelicoccus* (*Firmicutes*), *Catenibacterium* (*Erysipelotrichia*), *Lactobacillus* (*Bacilli*), *Bacillus* (*Bacilli*), *Lactococcus* (*Firmicutes*), and *Ruminococcaceae* UCG-014 (*Firmicutes*). Some of these genera include known spore formers.

We also included negative controls (PCR-grade water) in all batches of DNA extractions and measured each negative control with bacterial 16S rRNA gene targeted qPCR. Extraction negative controls were always below the limit of detection for the qPCR assay.

We now convey this information in the Supplemental Methods and via a newly-added figure (Figure S5).

At the moment of reviewing this manuscript, the underlying data were not accessible (BioProject, Github resource).

Response: We apologize that the GitHub resource was not available to the reviewers. We did upload our git repository, which comprises all Jupyter Notebooks associated with this work, to GitHub prior to submitting the manuscript for peer review; however, we failed to make the repository public. The repository should now be

publicly available at [https://github.com/leylabmpi/16S-arc vertebrate paper](https://github.com/leylabmpi/16S-arc Vertebrate paper). The BioProject is currently private and will be made public upon acceptance of the manuscript to a peer-reviewed journal.

Additional specific comments:

The discussion is somewhat lengthy compared to the introduction, which was kept very short.

Response: We agree with the reviewer that the introduction is brief, but we feel that the introduction provides a concise description of the necessary background information and citations to relevant literature in order for the reader to comprehend our work. We do not feel that expanding upon the Introduction is optimal, given the word count limitations (<https://www.nature.com/nmicrobiol/about/content>) and the expanded text for the Results and Discussion that has resulted from the additional analyses and discussion suggested by each reviewer.

Line 22: gastrointestinal samples ? (maybe explain in more detail afterwards what is the difference of fecal and gut samples)

Response: We thank the reviewer for suggesting this simpler term. We have edited the text in the Abstract and also clarify the “gastrointestinal” terminology in the Methods.

Line 26: Methanothermobacter should be written in italics, please check also elsewhere: all genus and species names should be written in italics

Response: We did not italicize the genus name in this case because it is not used as binomial nomenclature (*i.e.*, “*Genus species*” when specifically referring to a particular species); however, most journals seem to now require italicization of all taxonomic classifications (*e.g.*, “*Firmicutes*”, “*Methanothermobacter*”, and thus “*Bacteria*” when referring to the taxonomic domain; see <https://mbio.asm.org/sites/default/files/additional-assets/mbio-ITA.pdf> for an example), so we will follow this editorial style. We have italicized all taxonomic classifications throughout the manuscript.

Line 26: “enriched” is not correct here, as enrichment would mean actual numbers, it’s only relative information

Response: We have changed the term to “relatively abundant” to describe the higher proportional abundance of *Methanothermobacter* in *Aves* relative to other host clades.

Line 30: “common ancestor” (not command)

Response: We thank the reviewer for pointing out this error. We have corrected it.

Line 35: rephrase “waste products”

Response: We have rephrased this sentence in order to clarify.

Line 37: 16S rRNA GENE sequencing (please correct throughout); likewise, please correct: 16S rRNA gene primers

Response: We agree with the reviewer that “16S rRNA gene” is the correct terminology. However, due to word count constraints (<https://www.nature.com/nmicrobiol/about/content>), we clarify in the Introduction that “16S” stands for “16S rRNA gene” and use “16S” throughout the rest of the manuscript.

Line 74: please explain, what do you mean with “adequate sequence data”

Response: We refer to our sequence depth cutoff of 250 sequences per sample, which we describe in detail in the Methods. We have inserted a reference to the Methods in order to help clarify our meaning.

Fig. 1: How was the cladogram (diet) prepared? Colors should be adapted (captive/wild in 1A and 1D; Diet: 1A and 1C, etc...)

Response: The dendrogram in Figure 1A that describes diet similarity was constructed from UPGMA clustering of Jaccard similarities of diet component presence/absence among host species. We have expanded upon our existing description in the figure legend in order to clarify. We have also edited the colors in C and D in order to match those in A.

Line 109: more precise information?

Response: We have edited this sentence to provide more quantitative information on our qualitative statement of “taxonomic novelty to cultured representatives”.

Fig. 2: 2A consider to reduce information to one specific measure; Explain bracket without stars in Fig. 2E

Response: We believe that all measures provide uniquely useful information on community diversity that not one single diversity measure can provide. Assessing multiple beta diversity measures is a common practice among microbiome studies for this reason. We therefore feel that our analysis of four common beta diversity measures is appropriate in the context of this work.

We have clarified in the Figure 2 legend that the bracket lacking a star above it denotes a non-significant difference between the two groups.

Line 202: explain CI

Response: We have edited the text to denote CI as “confidence interval”.

Line 276: please explain in more detail what is shown in the sub-networks.

Response: We have clarified how the sub-networks were defined and also refer to the Methods, which provide more details about this approach.

In general, please expand a bit on the sampling of wild animals to give the reader more background information without having to read the previous work.

Response: We have expanded on the sample collection methods.

Also, please be careful when using “abundances” to state that you refer to relative abundances. (e.g. lines 216 ff)

Response: We feel that we have thoroughly described our dataset as originating from 16S rRNA gene next generation sequencing, and such data is well-established as compositional via many scientific articles on the topic (e.g., <https://doi.org/10.3389/fmicb.2017.02224>, <https://doi.org/10.1038/s41467-019-10656-5>, <https://doi.org/10.1186/s12866-017-1101-8>, and <https://doi.org/10.1093/gigascience/giz107>). Therefore, any reference that we make to “abundance” in the manuscript must refer to “relative abundance” due to the compositional nature of the data, and so we feel that the repeated use of “relative abundance” throughout the manuscript text is not necessary, especially given the word count limitations of Nature Microbiology articles (<https://www.nature.com/nmicrobiol/about/content>).

Fig. 3: the color code used in the trees should be explained in the figure itself.

Response: We have added figure keys for each of the 3 phylogeny plots in order to describe the circle colors and sizes on each node of the trees.

Line 245: maybe the finding that *Methanothermobacter* is associated with higher temperature is only due to its association to Aves?

Response: We agree with the reviewer. We state in the Discussion that while the data is consistent with the hypothesis that *Methanothermobacter* is prevalent and relatively abundant in *Aves* due to high host body temperature, we cannot decouple host body temperature from host evolutionary history. Regardless, we feel that the data is compelling and leads to a directly-testable hypothesis to be addressed in future work.

Line 248: which evidence do you mean?

Response: We were referring to the figures and tables referenced earlier in the paragraph. We have added references to these data in this sentence.

Line 252: This requires a bit more support, such as qPCR, reconstructed genome fragments...

Response: We have added both qPCR data (Figure S27) and reconstructed genome fragment data (Figure S28) supporting our initial findings that *Methanothermobacter* is prevalent and likely a persistent inhabitant of many avian species, as stated in Line 252 of the initial submission.

Line 265: in my opinion, the archaea-bacteria interaction/co-correlation could be analysed more deeply. Which archaea-bacteria co-occurrences did you observe (species, genera...)

Response: We have added a table of all significant *Archaea-Bacteria* co-occurrences, summarized at the genus level (species-level classifications are not consistently robust when based on a small region of the 16S rRNA gene; Table S9). We have updated the Results to include a brief summary of these data.

Discussion: Consider to include a limitations chapter.

Response: We have edited the final paragraph of the Discussion to more directly address the major limitations of this study, especially the mammalian bias of our dataset. We also state that more work is needed to elucidate the role of host body temperature on the vertebrate gut microbiome, and we address the fact that our methodology does not assess the potential host and microbial genetic factors that mediate our observed pattern of phyllosymbiosis and cophylogeny.

Line 408: Materials and Methods: Silva 119 is already quite old and a more up-to-date version should be used

Response: We apologize for this error in the text. The Silva 119 was originally used in an initial analysis of this work. We used Silva 132 for the work presented in the manuscript. We have corrected this typo in the Supplemental Methods.

Reviewer #2 (Remarks to the Author):

This paper is a nice examination of archaeal diversity across 110 vertebrate species. It is well written and the analyses are appropriate and effectively utilized. Although the paper is clearly very descriptive, given how little we know about archaea in vertebrate hosts, the data represent a nice contribution to the literature.

That said, I do have an important concern about the study with regard to sample size. The authors state that, after samples were removed due to failure to amplify and pass quality filtering standards, they included 185 samples from 110 host species into the analysis. While this is a large number of host species, these numbers indicate that there are very few species for which multiple individuals were sampled/analyzed. In fact, it seems the majority of host species are likely to be represented by a single individual. In my mind, this sampling distribution has the potential to invalidate many of the analyses, particularly those that relate to host phylogeny.

For example, for ancestral state reconstruction, can we really be sure if these microbes are present/absent in a host species if we've only sampled one individual? While none of the analyses go as far as the host species level, I am still concerned that analyses examining higher levels of host taxonomy could be biased by this low coverage sampling. An important question would be how robust the results are to reducing the dataset to one in which at least five individuals per host species are sampled? Even this seems like low coverage, but is much better than one or two individuals. That said, I am not sure if most of the analyses performed are even possible on a dataset like that since I think it would reduce the number of host taxa represented substantially...

Response: We agree with the reviewer that the differential representation of samples per host species can introduce biases, which is why we employed an analysis methodology that directly addressed this issue (please see our response to Reviewer 1). Specifically, for most analyses in our work, we performed each analysis 100 times, each with a random dataset subset in which only one sample per host species was used. We therefore utilized only 1 sample per species for each individual test and only considered the overall hypothesis test significant if at least 95% of the individual tests were significant (as explained in the Methods and the Figure 1 legend). This approach assesses the robustness of the test to intra-species variance and performs each test with an equal number of samples per host species ($n = 1$).

Specifically in regards to the lack of intra-species sample replicates for some species, we note that the broad taxonomic scale of our analyses allow us to leverage inter-species variation across the vertebrate phylogeny, much like in a regression-based analysis, one leverages similarities among data points along the continuous independent variable (e.g., distance along a transect). If we were to focus on particular, finely resolved vertebrate clades (e.g., primates or a clade within primates), more samples from species within that clade and/or intra-species replication would likely be needed to assess the variance within the focal clade.

We note that the major exception to our general subsampling approach was the ancestral state reconstruction (ASR) analysis, in which the method itself leverages intra-species variance. We had failed to assess how biases in the number of samples per host (most numerous among *Artiodactyla* species), along with loss of representation from PCR/sequencing failures, affected the ASR results. To address this deficit, we have tested the robustness of our ASR analyses by including all failed samples (taxon abundances set to zero for these samples) and subsampling to ≤ 3 samples per host species (Figure S22). This analysis revealed that our results are robust to PCR/sequencing failures and reducing the number of samples per host. We have updated the Results to include these new data.

We have clarified the point of our subsampling methodology in the last paragraph of the Discussion.

I was also interested in knowing more about the samples that did not amplify. Although I understand the authors' argument that samples that do not amplify or pass quality filtering could have either very low abundances of archaea or low quality DNA, it seems to me that they might be able to distinguish between these possibilities rather than vaguely speculating. Since the same samples were used for bacterial analyses in a previous paper, it would be possible to compare the number of samples that failed in both analyses and match samples to determine if the same samples failed both assays or not. For samples that failed both, it seems like that the DNA quantity/quality was low. For samples that passed the bacterial analysis but failed the archaeal analysis, it is more likely that the archaea are low abundance/less prevalent across the sample set. Doing this comparison and briefly discussing it (and/or altering any conclusions accordingly) would strengthen the paper in my opinion.

Response: We agree with the reviewer that more should be done to understand the reason why a large subset of samples failed PCR or sequence quality control, and whether those reasons are biological or technical. We have now assessed the potential association between sample success/failure and a number of possible factors not included in our original analysis (Figure S3). These factors include i) the amount of gastrointestinal material collected, ii) the concentration of gDNA extracted from each sample, iii) the gDNA 260:280 ratio, iv) microbial 16S rRNA gene copy numbers estimated via qPCR with "universal" 16S rRNA gene primers, and v) the ratio of Bacteria to Archaea as estimated via metagenome sequencing of the overlapping samples in the published work of Youngblut and colleagues (DOI: 10.1128/mSystems.01045-20). Of these factors, only 16S rRNA gene copy numbers significantly differed between the successful and failed samples, with higher copy numbers in the successful samples. These results suggest that microbial biomass is the major factor that determines success or failure. We have added these new data in the Results and discuss them in the first paragraph of the Discussion.

A few minor detailed comments from the text:

Line 169-171: It seems like this section or results should go above with the other alpha and beta diversity results despite relying on the same analysis described here.

Response: Given the complexity of our analysis methods, we believe that it is important for clarity to introduce one analysis method at a time. Moving these RRPP + PGLS results up to the sub-section describing the MRM results would require a detailed description of 3 complicated analyses in the same paragraph. We therefore feel that the current order in which we present these results provides more clarity than the proposed alternative.

Line 335: I am understanding that the authors are saying methanogens are less likely to vary with host diet here, but weren't methanogens flagged as the subset of ASVs being driven by diet earlier in the manuscript?

Response: We did indeed find that bacterial community diversity associated more with diet than the archaeal community, as a whole. However, this finding does not preclude the possibility that the abundance of some specific archaeal taxa is associated with diet. This point is stated later in this same paragraph: “Third, diet may only strongly influence a minority of archaeal ASVs, given that we only observed significant associations between ASV abundances and host diet for 15 ASVs (Figures 2B & S16).”

Methods: Information about sequencing depth overall and per sample would be helpful to include.

Response: We have added a supplemental table of the total archaeal sequence counts per sample (Table S3) and refer to this table in the first paragraph of the Results.

Reviewer #3 (Remarks to the Author):

Overall I love this paper, strong statistical analysis and compelling findings.

Ln 73-75 – it is possible this is due to variable sample size obtained from different animals? While it appears as though biological replication was performed, it was variable across the species tested, is this because replicates failed? Was any attempt made to associate failure with DNA concentration obtained, as normalized by biomass extracted?

Response: Please see our response to Review 2 on this matter. As in our response to Review 2, we note that we have added a new figure (Figure S3) and edited to Results and Discussion to include these new findings. We thank both reviewers for suggesting this analysis.

It is not exactly clear to me whether a single ASVs was observed to be shared between species, or whether such sharing was not observed? e.g. Batharchaeia ASVs were also distributed differently between different species, right?

Response: The same ASVs were often observed across samples. We have included a new supplemental figure showing the distribution of ASV prevalence across all samples and species (Figure S6) and remark on ASV prevalence and refer to this figure in the Results and Supplemental Results.

Most of the *Bathyarchaeia* ASVs were only found in 1 sample, but 2 *Bathyarchaeia* ASVs were found in both Smooth Newt samples (Figure S6). We now note this in the Supplemental Results.

“Of 140 samples that overlap between our Archaea-targeted 16S dataset (“16S-arc”) and that from our previous work with standard “universal” 16S primers (“16S-uni”), 1390 versus only 169 archaeal ASVs were observed in each respective dataset (Figure S7)” – is this just referring to the number of ASVs or the number of ASVs that overlap (is that even possible? i.e. did you see ASVs that were identical across the datasets with different primers? I couldn’t find that analysis in the supplementary methods, but I may have missed it).

Response: Due to the differences between the primer sets, we could not match ASV sequences across the sequence datasets. We were just referring to the number of ASVs per primer set across the overlapping samples for which both primer sets yield usable sequence data. We have clarified this in the figure legend (Figure S11).

Would have been nice to get the qPCR for these primers across all samples? Is that still possible? This influences whether you use ASV abundances (as used currently), versus ASV proportions, which is more appropriate here.

Response: Due to the lack of existing *Methanothermobacter*-specific qPCR primers, we developed our own primer design software and generated primers that target a core gene of the *Methanothermobacter* pan-genome (primers “174FR”). We used these primers to assess *Methanothermobacter* prevalence across avian and mammalian samples (Figure S27), and the data supported our initial hypothesis that *Methanothermobacter* is prevalent among avian and some mammalian taxa, and is likely a permanent inhabitant of many host species. We note that our qPCR assay could not be applied to all samples due to limited amounts of usable gDNA for many of the samples. Please see our response to Reviewer 1 for more details about the primer design, qPCR assay, and the resulting findings.

Fig S8 ordination plots – what explains the strange ordination for weighted matrices? It looks like there are two different gradients, and maybe each is driven by proportions of one sequence/taxon, driving the data in that bifurcated arrow like pattern?

Response: The “arch” pattern shown in the weighted beta-diversity PCoAs can occur when the species composition of the sites progressively changes along the environmental gradient (See Ramette 2007, <https://dx.doi.org/10.1111%2Fj.1574-6941.2007.00375.x>). In such an instance, the PCoA obtains axes that both i) maximally separate species and ii) are uncorrelated with the other axes, by “folding” the first axis in the middle and bringing its extremities together, thus resulting in an arch pattern. Certain detrending approaches can be used to mitigate this arch effect, but we do not feel such measures are necessary, given i) their added

complexity in implementation and data interpretation, and ii) that the PCoAs are only used for a qualitative assessment of the data to support our MRM analysis results.

Ln128-129 – did you run it for just mammalian species as well?

Response: We have now included an MRM analysis on just mammals (Figure S13). The variable with the strongest effect size for all beta diversity measures was still phylogeny, but it was not quite significant for both UniFrac measures, possibly due to the lower sample size and/or the effects of a broad cophylogeny signal spanning multiple vertebrate taxonomic classes. We have updated the Results subsection on the MRM results in

Decision Letter, first revision:

Dear Nick,

Thank you for your patience while your manuscript "Strong influence of vertebrate host phylogeny on gut archaeal diversity" was under peer-review at Nature Microbiology. It has now been seen by 3 referees, whose expertise and comments you will find at the of this email. You will see from their comments below that while they find your work of interest, some important points are raised. We are very interested in the possibility of publishing your study in Nature Microbiology, but would like to consider your response to these concerns in the form of a revised manuscript before we make a final decision on publication.

In particular, you will see that referee #1 has some remaining concerns with the Methanothermobacter mapping approach data and these should be thoroughly addressed. All three referees also ask that conclusions are further tempered in line with the data. The rest referees' reports are clear and the remaining issues should be straightforward to address.

We would also suggest that you reformat as an Analysis given the nature of the study. The format is very similar to an Article.

If you have not done so already please begin to revise your manuscript so that it conforms to our Analysis format instructions at <http://www.nature.com/nmicrobiol/info/final-submission/>. The main text (excluding abstract, online Methods, references and figure legends) is approximately 3,000. The abstract is typically 150 words, unreferenced. Analyses have no more than 6 display items (figures and/or tables). An introduction (without heading) is followed by sections headed Results, Discussion and online Methods. The Results and online Methods should be divided by topical subheadings; the Discussion does not contain subheadings. As a guideline, Analyses allow up to 25 references.

We have some flexibility, and can allow a revised manuscript at 3,500 words, but please consider this

a firm upper limit. There is a trade-off of ~250 words per display item, so if you need more space, you could move a Figure or Table to Supplementary Information.

Some reduction could be achieved by focusing any introductory material and moving it to the start of your opening 'bold' paragraph, whose function is to outline the background to your work, describe in a sentence your new observations, and explain your main conclusions. The discussion should also be limited. Methods should be described in a separate section following the discussion, we do not place a word limit on Methods.

Nature Microbiology titles should give a sense of the main new findings of a manuscript, and should not contain punctuation. Please keep in mind that we strongly discourage active verbs in titles, and that they should ideally fit within 90 characters each (including spaces).

Please include a data availability statement as a separate section after Methods but before references, under the heading "Data Availability". This section should inform readers about the availability of the data used to support the conclusions of your study. This information includes accession codes to public repositories (data banks for protein, DNA or RNA sequences, microarray, proteomics data etc...), references to source data published alongside the paper, unique identifiers such as URLs to data repository entries, or data set DOIs, and any other statement about data availability. At a minimum, you should include the following statement: "The data that support the findings of this study are available from the corresponding author upon request", mentioning any restrictions on availability. If DOIs are provided, we also strongly encourage including these in the Reference list (authors, title, publisher (repository name), identifier, year). For more guidance on how to write this section please see:

<http://www.nature.com/authors/policies/data/data-availability-statements-data-citations.pdf>

To improve the accessibility of your paper to readers from other research areas, please pay particular attention to the wording of the paper's opening bold paragraph, which serves both as an introduction and as a brief, non-technical summary in about 150 words. If, however, you require one or two extra sentences to explain your work clearly, please include them even if the paragraph is over-length as a result. The opening paragraph should not contain references. Because scientists from other sub-disciplines will be interested in your results and their implications, it is important to explain essential but specialised terms concisely. We suggest you show your summary paragraph to colleagues in other fields to uncover any problematic concepts.

If your paper is accepted for publication, we will edit your display items electronically so they conform to our house style and will reproduce clearly in print. If necessary, we will re-size figures to fit single or double column width. If your figures contain several parts, the parts should form a neat rectangle when assembled. Choosing the right electronic format at this stage will speed up the processing of your paper and give the best possible results in print. We would like the figures to be supplied as

vector files - EPS, PDF, AI or postscript (PS) file formats (not raster or bitmap files), preferably generated with vector-graphics software (Adobe Illustrator for example). Please try to ensure that all figures are non-flattened and fully editable. All images should be at least 300 dpi resolution (when figures are scaled to approximately the size that they are to be printed at) and in RGB colour format. Please do not submit Jpeg or flattened TIFF files. Please see also 'Guidelines for Electronic Submission of Figures' at the end of this letter for further detail.

Figure legends must provide a brief description of the figure and the symbols used, within 350 words, including definitions of any error bars employed in the figures.

Please include a statement before the acknowledgements naming the author to whom correspondence and requests for materials should be addressed.

Finally, we require authors to include a statement of their individual contributions to the paper -- such as experimental work, project planning, data analysis, etc. -- immediately after the acknowledgements. The statement should be short, and refer to authors by their initials. For details please see the Authorship section of our joint Editorial policies at http://www.nature.com/authors/editorial_policies/authorship.html

- * include a point-by-point response to any editorial suggestions and to our referees. Please include your response to the editorial suggestions in your cover letter, and please upload your response to the referees as a separate document.
- * ensure it complies with our format requirements for Letters as set out in our guide to authors at www.nature.com/nmicrobiol/info/gta/
- * state in a cover note the length of the text, methods and legends; the number of references; number and estimated final size of figures and tables
- * resubmit electronically if possible using the link below to access your home page:

{REDACTED}

*This url links to your confidential homepage and associated information about manuscripts you may have submitted or be reviewing for us. If you wish to forward this e-mail to co-authors, please delete this link to your homepage first.

Please ensure that all correspondence is marked with your Nature Microbiology reference number in the subject line.

Nature Microbiology is committed to improving transparency in authorship. As part of our efforts in this direction, we are now requesting that all authors identified as 'corresponding author' on published papers create and link their Open Researcher and Contributor Identifier (ORCID) with their account on the Manuscript Tracking System (MTS), prior to acceptance. This applies to primary research papers only. ORCID helps the scientific community achieve unambiguous attribution of all scholarly contributions. You can create and link your ORCID from the home page of the MTS by clicking on 'Modify my Springer Nature account'. For more information please visit www.springernature.com/orcid.

We hope to receive your revised paper within three weeks. If you cannot send it within this time, please let us know.

{REDACTED}

Reviewer Expertise:

Referee #1: archaea, sequencing

Referee #2: vertebrate microbiome, evolution

Referee #3: human microbiome, microbial ecology, sequencing

Reviewers Comments:

Reviewer #1 (Remarks to the Author):

Dear authors!

I appreciate your reponses and changes of the manuscript, which has improved substantially. I particularly appreciate the inclusion of additional experiments and analyses. Thank you very much, well done!

I only have a few remaining comments:

MINOR: Although I see the problem of the word count, maybe there is a bit of flexibility still to allow for the correct wording of 16S rRNA gene sequencing/primers instead of using the abbreviation. Same applies to the case of "relative abundance". In my opinion the correct (and lengthy) term needs to be

used to avoid misunderstandings.

MINOR: Controls: I strongly advise, for future studies, to sequence the negative controls and remove the contaminating reads from the entire dataset by e.g. decontam or similar tools. However: Thank you for including the controls information in the supplementary methods file.

MAJOR: Methanothermobacter qPCR: I cannot really follow the argument that the Methanothermobacter qPCR supports the ASV-based study. Fig. S27 does not show supportive evidence. In my opinion, the sentence in the discussion has to be changed to: "qPCR with Methanothermobacter-targeting primers and metagenome assembly only partially supported the ASV data".

Similarly, the wording of "While the pattern of Methanothermobacter dominance and prevalence among Aves was not as strong as observed with the ASV data" – it was not at all showing this, so "not as strong" is quite an embellishment ;-). I suggest to rephrase.

MAJOR: Where are the data of the Methanothermobacter mapping approach shown? Did you only map against Aves, or also mammals? Any differences here? This is important information to support your assumptions made, otherwise you need to adapt the conclusions made on the Methanothermobacter abundance. Also, I would like to stress, that even if you observe a change in relative abundances, the overall abundance might not necessarily change; this is however important biological information and should be reflected correctly in conclusions and discussion.

Again, thanks for the hard work, highly appreciated!

Reviewer #2 (Remarks to the Author):

The authors have fully responded to my comments. Overall, I am happy with their efforts and continue to think this is a nicely written, albeit descriptive, paper. I do still have concerns about within-host-species sample size. The authors have addressed these weaknesses to the best of their ability statistically and acknowledge this as a shortcoming in the manuscript as well. Beyond waiting for a more robust dataset, there is not much more that can be done. However, I would caution the authors to be a little bit more careful in some of their wording and conclusions. They should be explicit in the results about what level of host taxonomy 'phylogeny' refers to - it is explicitly stated later in the paper that class is the focus, but it is a little vague earlier in the paper. Also, there are results and conclusions that are described at the host species level still (e.g. in the discussion lines 310 and 345). It doesn't seem to me that this level of interpretation should be included.

Finally, am I misunderstanding or is the label of Figure 2A wrong? Is this truly 'intraspecies' or rather 'interspecies'?

Reviewer #4 (Remarks to the Author):

I have reviewed the manuscript by Youngblut and colleagues entitled "Strong influence of vertebrate host phylogeny on gut archaeal diversity", taking into account previous reviewer comments and the authors' responses. I would first like to state that this is a very interesting study and does indeed

expand on the paradigm of host-Archaea associations in animals.

I believe that the authors have done a sufficient job of addressing previous reviewer concerns, particularly with respect to their development and application of new primers for qPCR amplification of Methanothermobacter ASVs. This required substantial expansion upon their previous methods, and has the added benefit of providing the scientific community with some nice publicly available code. Additionally, the authors produced new sequence data from which they assembled Methanothermobacter contigs to provide further evidence for the presence of this archaea in their samples. I believe these efforts adequately address what I view as the only major former concern in the study.

Regarding limited intra-specific sampling, I believe that the authors statistical approach was sound, and their conclusions reasonable based on the statistical results.

A few minor issue I have would be the authors' adherence to what appears to be a pet hypothesis linking host body temperature to the presence and relative abundance of Methanothermobacter. While intriguing, support for this hypothesis disappears when taking host phylogeny into account. I do not disagree with some speculation on the relationship still being valid. However, I would suggest toning down the language a bit more than they have perhaps already done based on previous reviewer comments. I would also modify this statement in the abstract, as it is a bit misleading given the lack of statistically significant evidence when accounting for phylogeny (which is part of the title, and thus assumed to be factored into primary results stated in the abstract).

Lastly, I was disappointed not to see more discussion of the possible interactions of archaea, particularly Methanothermobacter, with bacterial taxa that are known to be abundant in the host gut. Particularly as the authors have published extensively on this topic. However, I understand that there is a limitation of space for this journal and that topic could take quite a few paragraphs to address.

Overall, I thoroughly enjoyed this manuscript, I think it is an excellent study and important topic for the field of microbiome research to consider, and I applaud the authors on making their code and analyses so clean and clearly accessible for the community.

Author Rebuttal, first revision:

Reviewer #1 (Remarks to the Author)

Major comments

Dear authors!

I appreciate your responses and changes of the manuscript, which has improved substantially. I particularly appreciate the inclusion of additional experiments and analyses. Thank you very much, well done!

Response: We thank the reviewer for the kind remarks. The additions to the manuscript required substantial effort, especially with developing novel software to produce suitable qPCR primers used to test our prediction regarding *Methanothermobacter* prevalence, but we believe the results were well worth the effort.

I only have a few remaining comments:

MINOR: Although I see the problem of the word count, maybe there is a bit of flexibility still to allow for the correct wording of 16S rRNA gene sequencing/primers instead of using the abbreviation. Same applies to the case of “relative abundance”. In my opinion the correct (and lengthy) term needs to be used to avoid misunderstandings.

Response: We have expanded such abbreviations in certain locations of the manuscript in order to help improve clarity. For example, we have edited the following sentence at the start of the Supplemental Results: “Of 311 genomic DNA samples from 5 vertebrate taxonomic classes, 185 (60%) passed 16S rRNA *gene* PCR amplification, MiSeq sequencing, and sequence data quality control (Table S2)”. We note that the supplemental text, which has no word limitations, generally includes unabbreviated versions of “16S rRNA gene sequencing/primers” and “relative abundance” (e.g., many of the subsection titles and table/figure captions).

MINOR: Controls: I strongly advise, for future studies, to sequence the negative controls and remove the contaminating reads from the entire dataset by e.g. decontam or similar tools. However: Thank you for including the controls information in the supplementary methods file.

Response: We agree with the reviewer that more effort could have been devoted to assessing the negative controls and clearly reporting our assessments. Still, we believe that the negative controls used in our work (including the actual sequencing and assessment of those negative controls) greatly helps to show the robustness of our methods and findings.

We note that the simple removal of all taxa observed in the negative controls can remove taxa truly present in the samples, especially given the lack of taxonomic resolution afforded by sequencing 1-2 variable regions of the 16S rRNA gene. We refer the reviewer to Edmonds and Williams 2018 (https://doi.org/10.1096/fasebj.31.1_supplement.940.3) and Nearing et al., 2021 (<https://doi.org/10.1186/s40168-021-01059-0>), which explore this dilemma in detail.

MAJOR: Methanothermobacter qPCR: I cannot really follow the argument that the Methanothermobacter qPCR supports the ASV-based study. Fig. S27 does not show supportive evidence. In my opinion, the sentence in the discussion has to be changed to: “qPCR with Methanothermobacter-targeting primers and metagenome assembly only partially supported the ASV data”. Similarly, the wording of “While the pattern of Methanothermobacter dominance and prevalence among Aves was not as strong as observed with the ASV data” – it was not at all showing this, so “not as strong” is quite an embellishment ;-). I suggest to rephrase.

Response: We agree with the reviewer that the qPCR data (and metagenome data) do not fully recapitulate the *Methanothermobacter* relative abundance pattern, as observed with the ASV data. This lack of coherence may have resulted from the limited amounts of gDNA available for the qPCR assay and a general lack of closely related genomic representatives for the metagenomics analysis. We note both of these possibilities in the Results. Another possible cause of the incongruence is the difference between absolute abundances, as measured via qPCR, and relative abundances derived from the ASV data. More detailed investigations of absolute abundances of all methanogens would help to clarify whether this is the main source of the incongruence.

We have modified the text to more clearly point out that the ASV and qPCR data do not match well. Specifically, we have modified the results to now state: “qPCR with novel *Methanothermobacter*-targeting primers *partially* supported our findings that *Methanothermobacter* is present among many avian and mammalian species (Figure S27)” and further along in the paragraph: “While *Methanothermobacter* was more sparsely observed among Aves via qPCR versus in the ASV data, this may have resulted from primer biases or lack of enough high quality gDNA for qPCR.” We have modified the Discussion to now state: “Moreover, qPCR with *Methanothermobacter*-targeting primers and metagenome assembly *partially* supported the ASV data by providing multiple lines of evidence that *Methanothermobacter* is present in many avian species (Figures S26, S27, & S28)”

MAJOR: Where are the data of the Methanothermobacter mapping approach shown? Did you only map against Aves, or also mammals? Any differences here? This is important information to support your assumptions made, otherwise you need to adapt the conclusions made on the Methanothermobacter abundance. Also, I would like to stress, that even if you observe a change in relative abundances, the overall abundance might not necessarily change; this is however important biological information and should be reflected correctly in conclusions and discussion.

Response: We thank the reviewer for pointing out that we failed to provide a reference to Figure S28 in the paragraph in the Results that describes these data. We have now added a reference to Figure S28: “The number of contigs that mapped to *Methanothermobacteraceae* or *Methanothermobacteraceae A* was 88 ± 799 (s.d.), which was $0.88\% \pm 0.93$ (s.d.) of all contigs assembled (Figure S28).”

Our focus of this analysis was to support our prediction that *Methanothermobacter* is present among *Aves* species. Therefore, we focused our NGS costs, computational effort, and analytical effort on addressing this particular research goal. We believe that extending this analysis to mammals is not needed to address whether *Methanothermobacter* is present among the *Aves*, especially given the added non-trivial computational costs and the increased complexity of the manuscript. As stated above, we have toned down the language in the Discussion to reflect what can be robustly concluded from our data: “Moreover, qPCR with *Methanothermobacter*-targeting primers and metagenome assembly *partially* supported the ASV data by providing multiple lines of evidence that *Methanothermobacter* is present in many avian species (Figures S26, S27, & S28)”

Again, thanks for the hard work, highly appreciated!

Response: Thank you for helping to improve this work!

Reviewer #2 (Remarks to the Author):

The authors have fully responded to my comments. Overall, I am happy with their efforts and continue to think this is a nicely written, albeit descriptive, paper. I do still have concerns about within-host-

species sample size. The authors have addressed these weaknesses to the best of their ability statistically and acknowledge this as a shortcoming in the manuscript as well.

Response: We thank the reviewer for the supportive comments. In regards to the variability of samples per host species, we note that this is a pervasive problem in the state-of-the-art for this field. For example, here are some notable recent studies that assessed gut/skin microbiome diversity in large sets of vertebrates: Levin et al., 2021. Science (1 - 5 samples per species), Kartzinel et al., 2019. PNAS (1 - 29 samples per species), Xiao et al., 2021. mSphere (1 - 10 samples per species), Milani et al., 2020. ASM (1 - 19 samples per species), Amato et al., 2019. Genome Biology (3 - 10 samples per species), Moeller et al., 2017. PNAS (1 - 34 samples per species), Woodhams et al., 2020. Genome Biology (1 - 1856 samples per species; mean = 24), Ross et al., 2018. PNAS (1 - 77 samples per species). The logistics of sample collection greatly hinder the task of obtaining a consistent number of samples per species, even for studies assessing captive animals in zoos (e.g., Xiao et al., 2021. mSphere). Nevertheless, we will strive for more balanced sampling across host species in future work and apply statistical methodology that best deals with such imbalances when they do manifest.

Beyond waiting for a more robust dataset, there is not much more that can be done. However, I would caution the authors to be a little bit more careful in some of their wording and conclusions. They should be explicit in the results about what level of host taxonomy 'phylogeny' refers to - it is explicitly stated later in the paper that class is the focus, but it is a little vague earlier in the paper.

Response: We use “phylogeny” in the paper only to refer to the host phylogeny, and not the taxonomy (e.g., class level). We do summarize some data by host taxonomic groupings (e.g., class level), but as stated in the methods, we used the phylogeny (continuous variable) instead of taxonomic groupings (discrete variable) for hypothesis testing (e.g., Figure 2A). Utilizing a continuous variable for regression-based analyses alleviates issues associated with imbalanced categories (e.g., number of samples per species).

Also, there are results and conclusions that are described at the host species level still (e.g. in the discussion lines 310 and 345). It doesn't seem to me that this level of interpretation should be included.

Response: Lines 310 & 345 simply describe the distribution of certain microbial taxa among vertebrate hosts, which allows the reader to conjecture about possible reasons for the observed distributions. We do not make claims of robust microbial diversity patterns based on these simple observations.

Finally, am I misunderstanding or is the label of Figure 2A wrong? Is this truly 'intraspecies' or rather 'interspecies'?

Response: “Intra-species” is correct in this context. As stated in the methods and figure caption, the analysis involved repeated hypothesis testing with randomly sampled subsets of the data (one sample per species for each subset). The boxplots display the variance in effect sizes observed among those randomly sampled subsets; therefore, the variance is describing the variance within samples from the same species. We have clarified this in the figure caption by adding the following sentence: “The boxplots show the variance in effect sizes observed among each dataset permutation ($n = 100$).”

Reviewer #4 (Remarks to the Author):

I have reviewed the manuscript by Youngblut and colleagues entitled "Strong influence of vertebrate host phylogeny on gut archaeal diversity", taking into account previous reviewer comments and the authors' responses. I would first like to state that this is a very interesting study and does indeed expand on the paradigm of host-Archaea associations in animals.

I believe that the authors have done a sufficient job of addressing previous reviewer concerns, particularly with respect to their development and application of new primers for qPCR amplification of Methanothermobacter ASVs. This required substantial expansion upon their previous methods, and has the added benefit of providing the scientific community with some nice publicly available code. Additionally, the authors produced new sequence data from which they assembled Methanothermobacter contigs to provide further evidence for the presence of this archaea in their samples. I believe these efforts adequately address what I view as the only major former concern in the study.

Response: We thank the reviewer for the generous remarks.

Regarding limited intra-specific sampling, I believe that the authors statistical approach was sound, and their conclusions reasonable based on the statistical results.

Response: We thank the reviewer for the supportive comments. Animal gut microbiome data analysis is challenging, especially due to intra-specific variance and imbalances in number of samples per species (see our response to Reviewer 2). We have striven to utilize and develop best practices as the field advances.

A few minor issue I have would be the authors' adherence to what appears to be a pet hypothesis linking host body temperature to the presence and relative abundance of *Methanothermobacter*. While intriguing, support for this hypothesis disappears when taking host phylogeny into account. I do not disagree with some speculation on the relationship still being valid. However, I would suggest toning down the language a bit more than they have perhaps already done based on previous reviewer comments. I would also modify this statement in the abstract, as it is a bit misleading given the lack of statistically significant evidence when accounting for phylogeny (which is part of the title, and thus assumed to be factored into primary results stated in the abstract).

Response: We have toned down the wording on this putative association between *Methanothermobacter* abundance and body temperature. Specifically, we have changed the line in the Discussion that stated: “These assemblages reflect the different distributions of each methanogen genus, which is likely influenced by body temperature (Figures S24 & S25)” to the following: “These assemblages reflect the different distributions of each methanogen genus, which *may be* influenced by body temperature (Figures S24 & S25)”. In the Abstract, we clarify that the association between *Methanothermobacter* abundance and host body temperature could not be decoupled from host evolutionary history. The abstract now states: “We provide evidence for novel *Archaea*-host associations, including *Bathyarchaeia* and *Methanothermobacter* — the latter of which was prevalent among *Aves* and relatively abundant in species with higher body temperatures, although this association could not be decoupled from phylogeny.”

Lastly, I was disappointed not to see more discussion of the possible interactions of archaea, particularly *Methanothermobacter*, with bacterial taxa that are known to be abundant in the host gut. Particularly

as the authors have published extensively on this topic. However, I understand that there is a limitation of space for this journal and that topic could take quite a few paragraphs to address.

Response: We agree with the reviewer that this fascinating aspect of our findings deserves more attention, but we also agree that a thorough discussion on this single aspect of this large and complex study would require a great deal of space. Moreover, the lack of additional data on these putative microbe-microbe associations (e.g., via direct co-culturing) would limit such a discussion mainly to conjecture about the underpinning causes of the observed patterns. We are currently working toward investigating such interactions via multiple approaches, including pan-genomic and evolutionary approaches, along with direct cultivation.

Overall, I thoroughly enjoyed this manuscript, I think it is an excellent study and important topic for the field of microbiome research to consider, and I applaud the authors on making their code and analyses so clean and clearly accessible for the community.

Response: We thank the reviewer for these generous comments. We strive to make microbiome research as reproducible as possible, and we believe that providing the code used for data analysis is a crucial component of that goal. We hope that the primer design software will be found as a generally useful tool for the microbiome research community (<https://github.com/leylabmpi/CoreGenomePrimers>); we have found it particularly useful for our ongoing research.

Decision Letter, second revision:

Dear Nick,

Thank you for submitting your revised manuscript "Strong influence of vertebrate host phylogeny on gut archaeal diversity" (NMICROBIOL-20113585B). It has now been seen by the original referees and their comments are below. The reviewers find that the paper has improved in revision, and therefore we'll be happy in principle to publish it in Nature Microbiology, pending minor revisions to satisfy the referees' final requests and to comply with our editorial and formatting guidelines.

We are now performing detailed checks on your paper and will send you a checklist detailing our

editorial and formatting requirements in about a week. Please do not upload the final materials and make any revisions until you receive this additional information from us.

Thank you again for your interest in Nature Microbiology Please do not hesitate to contact me if you have any questions.

{REDACTED}

Decision Letter, Final checks:

Dear Nick,

Thank you for your patience as we've prepared the guidelines for final submission of your Nature Microbiology manuscript, "Strong influence of vertebrate host phylogeny on gut archaeal diversity" (NMICROBIOL-20113585B). Please carefully follow the step-by-step instructions provided in the attached file, and add a response in each row of the table to indicate the changes that you have made. Please also check and comment on any additional marked-up edits we have proposed within the text. Ensuring that each point is addressed will help to ensure that your revised manuscript can be swiftly handed over to our production team.

In recognition of the time and expertise our reviewers provide to Nature Microbiology's editorial process, we would like to formally acknowledge their contribution to the external peer review of your manuscript entitled "Strong influence of vertebrate host phylogeny on gut archaeal diversity". For those reviewers who give their assent, we will be publishing their names alongside the published article.

Nature Microbiology offers a Transparent Peer Review option for new original research manuscripts submitted after December 1st, 2019. As part of this initiative, we encourage our authors to support increased transparency into the peer review process by agreeing to have the reviewer comments, author rebuttal letters, and editorial decision letters published as a Supplementary item. When you submit your final files please clearly state in your cover letter whether or not you would like to participate in this initiative. Please note that failure to state your preference will result in delays in accepting your manuscript for publication.

Cover suggestions

As you prepare your final files we encourage you to consider whether you have any images or illustrations that may be appropriate for use on the cover of Nature Microbiology.

Nature Microbiology has now transitioned to a unified Rights Collection system which will allow our Author Services team to quickly and easily collect the rights and permissions required to publish your work. Approximately 10 days after your paper is formally accepted, you will receive an email in providing you with a link to complete the grant of rights. If your paper is eligible for Open Access, our Author Services team will also be in touch regarding any additional information that may be required to arrange payment for your article.

Please note that *Nature Microbiology* is a Transformative Journal (TJ). Authors may publish their research with us through the traditional subscription access route or make their paper immediately open access through payment of an article-processing charge (APC). Authors will not be required to make a final decision about access to their article until it has been accepted. [Find out more about Transformative Journals](https://www.springernature.com/gp/open-research/transformative-journals)

Authors may need to take specific actions to achieve compliance with funder and institutional open access mandates. For submissions from January 2021, if your research is supported by a funder that requires immediate open access (e.g. according to [Plan S principles](https://www.springernature.com/gp/open-research/plan-s-compliance)) then you should select the gold OA route, and we will direct you to the compliant route where possible. For authors selecting the subscription publication route our standard licensing terms will need to be accepted, including our [self-archiving policies](https://www.springernature.com/gp/open-research/policies/journal-policies). Those standard licensing terms will supersede any other terms that the author or any third party may assert apply to any version of the manuscript.

For information regarding our different publishing models please see our page

[href="https://www.springernature.com/gp/open-research/transformative-journals">](https://www.springernature.com/gp/open-research/transformative-journals) Transformative Journals  page. If you have any questions about costs, Open Access requirements, or our legal forms, please contact ASJournals@springernature.com.

{REDACTED}

{REDACTED}

Final Decision Letter:

Dear Nick,

I am pleased to accept your Article "Vertebrate host phylogeny influences gut archaeal diversity" for publication in Nature Microbiology. Thank you for having chosen to submit your work to us and many congratulations.

Before your manuscript is typeset, we will edit the text to ensure it is intelligible to our wide readership and conforms to house style. We look particularly carefully at the titles of all papers to ensure that they are relatively brief and understandable.

Acceptance of your manuscript is conditional on all authors' agreement with our publication policies (see www.nature.com/nmicrobiolate/authors/gta/content-type/index.html). In particular your manuscript must not be published elsewhere and there must be no announcement of the work to any media outlet until the publication date (the day on which it is uploaded onto our website).

Please note that *Nature Microbiology* is a Transformative Journal (TJ). Authors may publish their research with us through the traditional subscription access route or make their paper immediately open access through payment of an article-processing charge (APC). Authors will not be required to make a final decision about access to their article until it has been accepted. [Find out more about Transformative Journals](https://www.springernature.com/gp/open-research/transformative-journals)

Authors may need to take specific actions to achieve [compliance](https://www.springernature.com/gp/open-research/funding/policy-compliance-faqs) with funder and institutional open access mandates. For submissions from January 2021, if your research is supported by a funder that requires immediate open access (e.g.

according to [Plan S principles](https://www.springernature.com/gp/open-research/plan-s-compliance)) then you should select the gold OA route, and we will direct you to the compliant route where possible. For authors selecting the subscription publication route our standard licensing terms will need to be accepted, including our [self-archiving policies](https://www.springernature.com/gp/open-research/policies/journal-policies). Those standard licensing terms will supersede any other terms that the author or any third party may assert apply to any version of the manuscript.
